# MMEvol: Empowering Multimodal Large Language Models with Evol-Instruct

## Abstract

The development of Multimodal Large Language Models (MLLMs) has seen significant advancements with increasing demands in various fields (e.g., multimodal agents, embodied intelligence). While model-driven approaches attempt to enhance MLLMs capabilities through diverse architectures, the gains have become increasingly marginal. Conversely, data-driven methods, which scale up image-text instruction data, are more effective but face limited data diversity and complexity challenges. The absence of high-quality data constitutes a significant development barrier for MLLMs. To address the data quality bottleneck, we propose **MMEvol**, a novel multimodal instruction data evolution framework. This framework iteratively improve data quality through a refined combination of fine-grained perception, cognitive reasoning, and interaction evolution, generating a more complex and diverse image-text instruction dataset that empowers MLLMs with enhanced capabilities. Beginning with an initial set of instructions, SEED-163K, we utilize **MMEvol** to systematically broaden the diversity of instruction types, extend visual reasoning steps to improve cognitive reasoning abilities, and thoroughly explore fine-grained information within images to enhance visual understanding and robustness. To comprehensively evaluate the effectiveness of our approach, we conduct extensive qualitative analysis and quantitative experiments across 13 vision-language tasks. Compared to baseline models trained with the initial seed data, the results demonstrate that our method achieves an average accuracy improvement of 3.1 percentage points. Furthermore, our approach reaches state-of-the-art (SOTA) performance in nine tasks using significantly less data compared to state-of-the-art models.

## 1 Introduction

*"The True Acquisition of Knowledge Lies in Grasping the Most Subtle Details."*

*Aristotle, circa 4th century BCE*

Multimodal Large Language Models (MLLMs) (Liu et al., 2024b;a; Li et al., 2023b; Dong et al., 2023; Sun et al., 2023b; Dai et al., 2024; Luo et al., 2024; Qi et al., 2024) have seen rapid development over the past two years and have become the preferred approach for various vision-language tasks (Kembhavi et al., 2016; Fu et al., 2024; Zhang et al., 2024a; Qian et al., 2024). By aligning visual encoders (Radford et al., 2021; Zhai et al., 2023; Sun et al., 2023a) with LLMs (Touvron et al., 2023; Bai et al., 2023; Lu et al., 2024; Young et al., 2024; Tao et al., 2024), and employing large-scale coarse-grained image-text pre-training (Zhu et al., 2024; Schuhmann et al., 2022; 2021) followed by small-scale instruction-tuning (Chen et al., 2024a; Liu et al., 2024b), MLLMs have demonstrated impressive capabilities across numerous vision-language tasks and are widely applied in many domains (e.g., multimodal agents, embodied intelligence). Model-driven approaches (Luo et al., 2024; Liu et al., 2024a; Tong et al., 2024; Zhang et al., 2024b) aim to integrate knowledge from images and text more efficiently by designing different network architectures to improve MLLMs performance. However, their effectiveness is diminishing due to the lack of high-quality data, and redundant model designs struggle to realize their potential and push the boundaries of model intelligence. In contrast, data-driven methods (Liu et al., 2024b; Chen et al., 2024a; Yu et al., 2023; Liu et al., 2024c; Fang et al., 2024; Chen et al., 2023), while more effective, often produce data with limited diversity and complexity. The lack of high-quality data remains a significant obstacle in the development of MLLMs. Therefore, the need of developing automated methods that can generate

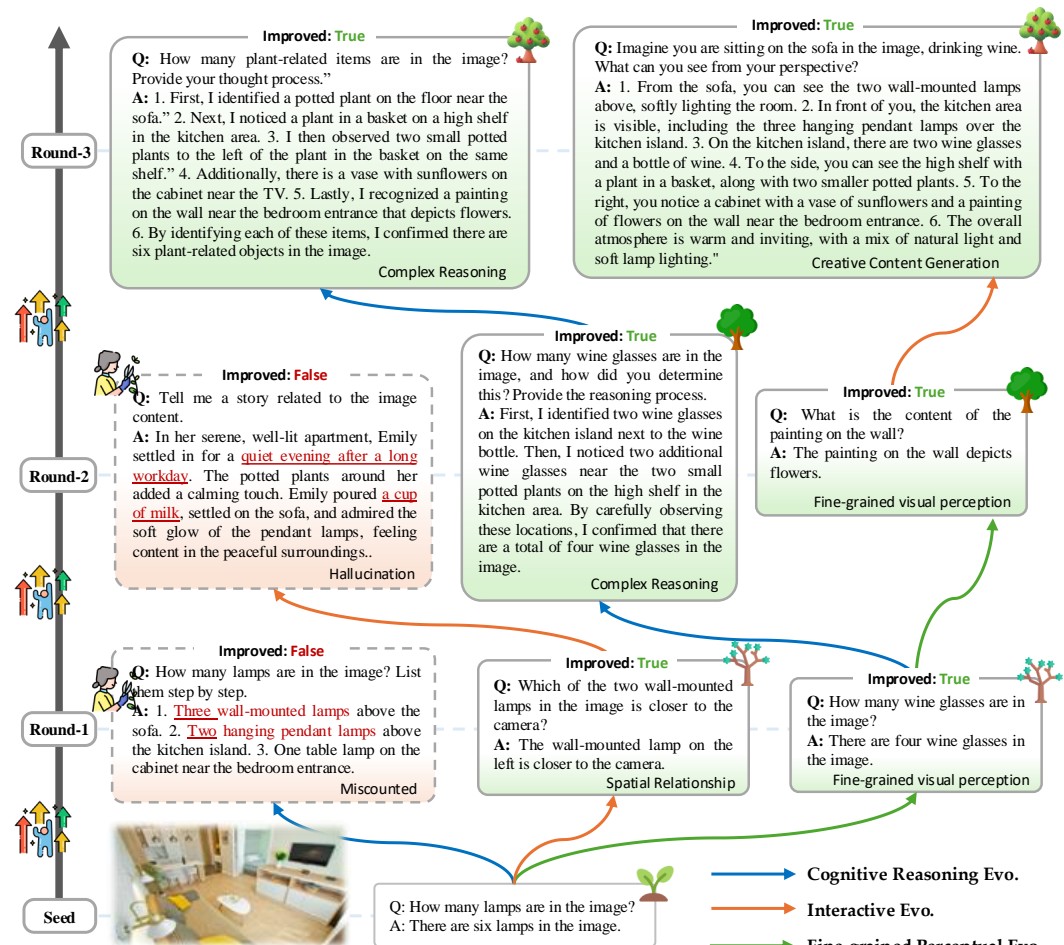

Figure 1: **Overview of MMEvol.** Instruction evolution and instruction elimination synergistically collaborate through multiple rounds to enhance the diversity and complexity of instruction data.

more challenging and diverse instructional data at a relatively low cost is urgent for empowering MLLMs with enhanced capabilities.

Analysis of existing data-driven methods for generating image-text instruction data reveals three common limitations: 1) **Limited instruction diversity**. Manually annotated instructions are constrained by the cognitive limitations of annotators, while model-generated instructions are limited by template presets, making it difficult to meet the diverse task requirements of the real world. This restricts the instruction-following ability of MLLMs. 2) **Limited instruction complexity**. Manual annotations often result in instructions of simple or moderate complexity, and automatically generated instructions tend to be brief and lacking in visual reasoning steps, which limits the model's ability to handle complex tasks. 3) **Insufficient alignment granularity**. Both manually and model-generated instructions primarily focus on common objects, neglecting rare or small objects, resulting in limited granularity in image-text alignment. This affects the model's visual perception robustness and resistance to hallucinations.

To address these limitations, we propose **MMEvol**, a novel method that utilizes advanced MLLMs for iterative evolution. This method automatically generates various types of open-domain instructions on a large scale, covering different difficulty levels to enhance the performance of MLLMs. Given that visual-language instruction data are constrained by visual content, the data generated through multiple iterations with Evol-Instruct (Xu et al., 2023; Luo et al., 2023a;b) tend to include simple restatements and data unrelated to visual content, making deep and broad evolution challenging. Therefore, we have made several adjustments to the evolution prompting process, ultimately developing an image-text instruction evolution paradigm. These adjustments include a more refined image-text instruction data paradigm and the definition of three evolution directions: fine-grained perception

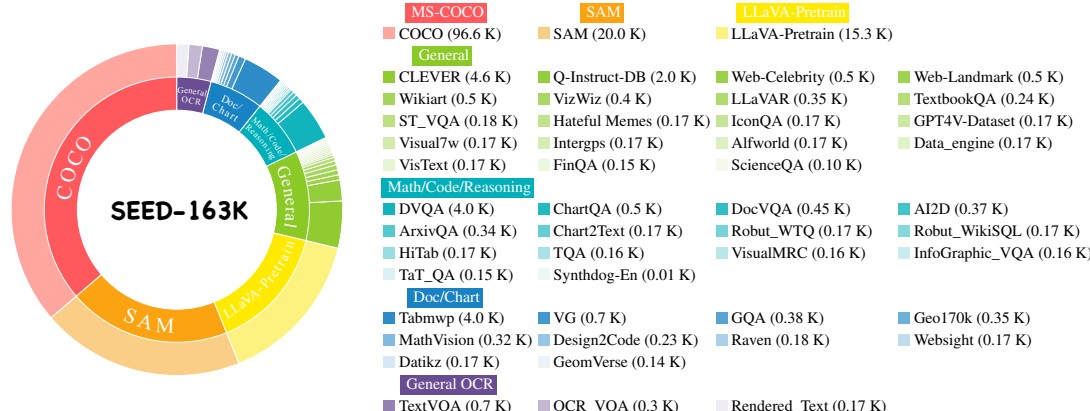

Figure 2: **SEED-163K: 163K Curated Seed Instruction Tuning Dataset for Evol-Instruct. Left:** The inner circle shows the original distribution of SEED-163K. The outer circle shows the curated SEED-163K. **Right:** All the data sources in the SEED-163K dataset, as well as the ones filtered in data curation.

evolution, cognitive reasoning evolution, and interaction evolution. The **MMEvol** mechanism is summarized in Fig. 1, with each evolution cycle comprising two main steps: instruction evolution and instruction elimination. Instruction evolution randomly selects one of fine-grained perception evolution, cognitive reasoning evolution, or interaction evolution, upgrading simple instructions to more complex or diverse ones. Specifically, fine-grained perception evolution aims to leverage visual information in images to generate data with more detailed information; cognitive reasoning evolution prolongs the visual operation reasoning steps of instructions to increase their complexity; and interaction evolution aims to enhance instruction diversity by providing a wider variety of instruction forms. To account for occasional failures in evolved instructions, we use instruction elimination to filter out failed evolution. **MMEvol** repeats the instruction evolution and elimination processes multiple times to obtain a complex instruction dataset containing various instruction forms.

To validate the effectiveness of **MMEvol**, we perform three rounds of evolutionary iterations on 163K seed data, leading to 447K evolved samples. We fine-tuned the open-source LLaVA-NeXT (Liu et al., 2024a) model with these evolved data and compared it with other advanced methods across 13 vision-language benchmarks. Our method achieves state-of-the-art (SOTA) performance, demonstrating the effectiveness and efficiency of **MMEvol**. Additionally, we conduct detailed qualitative analysis and ablation experiments to showcase the contribution of each component of our method. We hope that the released evolutionary data and code will assist the community in understanding that using a small amount of high-quality image-text instruction data is far more critical than training MLLMs with large-scale low-quality image-text instruction data.

Our main contributions can be summarized as follows:

- A image-text instruction evolution framework, **MMEvol**, is designed to leverage advanced MLLMs, automating the generation of open-domain image-text instruction data across varying difficulty levels to enhance the diversity and complexity of existing datasets.

- By utilizing instruction evolution data, a high-quality data recipe is composed, and the evolved data will be released to advance the capabilities of other open-source MLLMs further.

- We train an MLLM using this high-quality data recipe, achieving superior performance in various downstream visual-language tasks compared to other fully open-source methods.

- The effectiveness and efficiency of the proposed approach are validated through extensive qualitative and quantitative analyses.

## 2 METHOD

In this section, we first introduce the curation of seed instruction data and then elaborate on the methodological details of **MMEvol**. Due to the space limitation, we simplify the seed data curation process and prompt templates. More details can be found in the Appendix E.

## Context Type I: Caption

The image shows a modern living room with natural light streaming through a large window... A black couch against a gray wall, ..., a glass coffee table that holds a white vase and a plant... The table rests on a beige rug, contrasting with the hardwood floor, adding warmth. The design suggests a comfortable and stylish living area.

## Context Type II: Visual Object Locations

window : [0.2 0.23 0.57 0.4], couch : [0.17 0.43 0.83 0.79], vase : [0.5 0.51 0.58 0.72] ...

----------------------------------------

## Vision/Language-Centered Multimodal Atomic Propositions & Permitted Vision-Centric Manipulations

Grounding Ability -> Grounding_i(tgt)->bbx_
Referencing Ability -> Referring_i(bbx)->tgt_i
Calculating Ability -> Calculate(tgt)->res_i
OCR Ability -> OCR_i(tgt)->txt_i
Existence Ability

Relationship Description Ability
Context Understanding Ability
Behavior Prediction Ability
Knowledge Integration Ability

## Objective

...

----------------------------------------

## Given Q&A

```
{
    "objects" : [window, couch, vase, ...]
    "skills" : ["Grounding Ability", "Context Understanding Ability", ...]
    "format" : "Complex reasoning",
    "question": "Which is closer to the window, couch or vase?"
    "steps": [{"manipulation": "grounding_1(`window`)-> bbx_1", "description": "Locate the window in the room
and return its bounding box as `bbx_1`".}, {"manipulation": "grounding_2(`couch`)-> bbx_2", "description":
"Locate the couch in the room and return its bounding box as ` bbx_2"}, {"manipulation": "grounding_3(`vase`)->
bbx_3", "description": "Locate the vase in the room and return its bounding box as `bbx_1`"}, {"manipulation":
"Calculate([window, couch/vase])->res_i", "description": "Calculate the spatial distance between window and
couch/vase"}],
    "answer" : First, based on the detected bounding boxes of the window [0.2, 0.23, 0.57, 0.4] and the couch [0.17,
0.43, 0.83, 0.79], it can be inferred that the couch is positioned below the window. Further, according to the
location of the vase [0.5 0.51 0.58 0.72], it is located near the center of the room. Therefore, the couch is closer to
the window.
}
```

## Rewritten Q&A

Figure 3: **Prompt Head of MMEvol.** The top block showcases the contexts such as caption and visual object locations, and the middle block demonstrates vision/la nguage-centered atomic propositions and evolution objective (described later). Additionally, we endow vision capabilities with pseudo-function calls to enhance visual reasoning during evolutionary processes. Finally, the bottom block further elucidates the organized seed sample, which is subsequently sent to the MLLM for rewriting.

### 2.1 SEED DATA CURATION

The seed instruction data are curated from LLaVA-Instruct (Liu et al., 2024b) and ShareGPT4V (Chen et al., 2023) datasets, supplemented with additional scientific and chart data sampled from Cambrain-1 (Tong et al., 2024). This process involved careful selection and refinement to ensure the quality and diversity of the instructions. For instructions with only captions, we use the OpenAI GPT-4o mini API to generate seed instruction data. Ultimately, after merging and filtering, we obtained a comprehensive dataset consisting of 163K instruction samples with unique images, which serve as the foundation for our subsequent Evol-Instruct. The seed data mixture is shown in Fig. 2. Please refer to Appendix B for more details.

### 2.2 METHODOLOGICAL DETAILS

The evolution of image-text instruction data is constrained by visual information, requiring evolved instruction data to be relevant to image content to avoid hallucinations. This makes the diversity evolution of image-text instructions particularly challenging. Additionally, the complexity evolution

> ## Objective of Fine-grained Perceptual Evolution
>
> I want you act as a Q&A Creator. Your objective is to draw inspiration from the given Q&A to create a brand new created Q&A. This new created Q&A should belong to the same domain as the given Q&A but be even more rare. The difficulty level of the created Q&A should be similar to that of the given Q&A. Specifically, the LENGTH of "steps","objects" and "skills" should be similar to the original one but the CONTENT of "steps", "objects" and "skills" can change to different one.

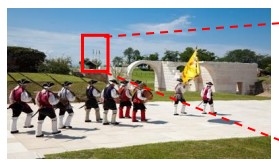 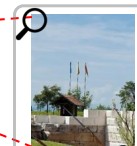 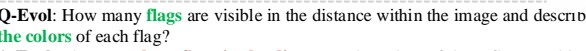

**Q**: How many people are there in the image?
**A**: There are nine people in the image.

**Q-Evol**: How many **flags** are visible in the distance within the image and describe **the colors** of each flag?
**A-Evol**: There are **three flags in the distance**. The colors of these flags are blue, yellow and red, respectively.

Figure 4: **Fine-grained perceptual evolution prompt and example.** Fine-grained perceptual evolution can generate samples with more detailed visual information, enhancing data diversity, which are marked with different colors for better visualization.

> ## Objective of Cognitive Reasoning Evolution
>
> I want you act as a Q&A Rewriter. Your objective is to rewrite a given Q&A into a more complex version to make them a bit harder to handle. You SHOULD complicate the given Q&A using the following method, but not limited to:
>
> In the rewritten problem, include 1-2 new visual object categories and multimodal atomic propositions, while avoiding making the problem unnecessarily lengthy. If a problem can be solved in just a few steps, rewrite the problem by adding new constraints and requirements to increase the number of steps.

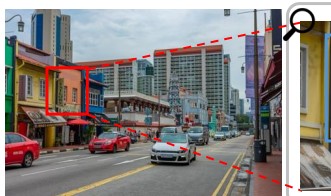 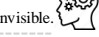

**Q**: What is the texts on the signboards?
**A**: "TAILORS", "254", "HAPPY HARRY'S" and "LORS" while others are invisible.

**Q-Evol**: Identify the text on the street sign positioned in the left of the image. **Let's consider the details step by step.**
**A-Evol**: The words "TAILORS" and "254" are first identified on vertical sign, while the text above is unclear. Another sign behind a signal board displayed "HAPPY HARRY'S", "LORS", and "254". Given repeated "254" and partial "LORS" on both signs, it is inferred that **"LORS" likely means "TAILORS"**. Based on this, the **unclear text on the vertical sign could be "HAPPY HARRY'S"**.

Figure 5: **Cognitive reasoning evolution prompt template and example.** Cognitive reasoning evolution can endow instruction data with a longer visual reasoning chain, increasing the complexity of the data. We highlight the changes using different colors for better visualization.

process of image-text instruction data often results in shallow reasoning phenomena, with MLLMs struggling to provide complex answers. As shown in Fig. 1, to address these issues and improve the success rate of evolution, we include carefully designed domains such as visual objects, atomic capabilities, visual manipulations, and instruction formats to standardize each instruction data format. The visual object domain includes visual objects in the images involved in the instruction data, implicitly constraining the evolution data and reducing visual hallucinations. We also summarize nine types of atomic capabilities involved in image-text instruction data to populate the atomic capability domain, aiming to enhance data diversity. Specifically, this includes five vision-centric capabilities: localization, reference, computation, optical character recognition (OCR), and existence judgment, and four language-centric capabilities: relation description, scene understanding, behavior prediction, and world knowledge association. The visual manipulation domain includes visual manipulation chains for problem-solving, where each step of the visual manipulation is based on vision-centric atomic capabilities, explicitly defining the visual reasoning process to mitigate shallow reasoning. The instruction format domain specifies the interaction types of the instruction data. These adaptations enhance the diversity and complexity of image-text instruction data and improve the success rate of evolution.

**Fine-grained Perceptual Evolution.** The goal of fine-grained perceptual evolution is to maximize the extraction of available visual information from images, especially overlooked non-primary visual objects. We observe that most instruction data tend to construct questions involving primary objects in images while neglecting less frequent non-primary objects. This results in a lack of instructions related to long-tail distribution objects. Training with such data can lead to visual hallucinations and poor generalization and robustness. Fine-grained perceptual evolution generates questions involving

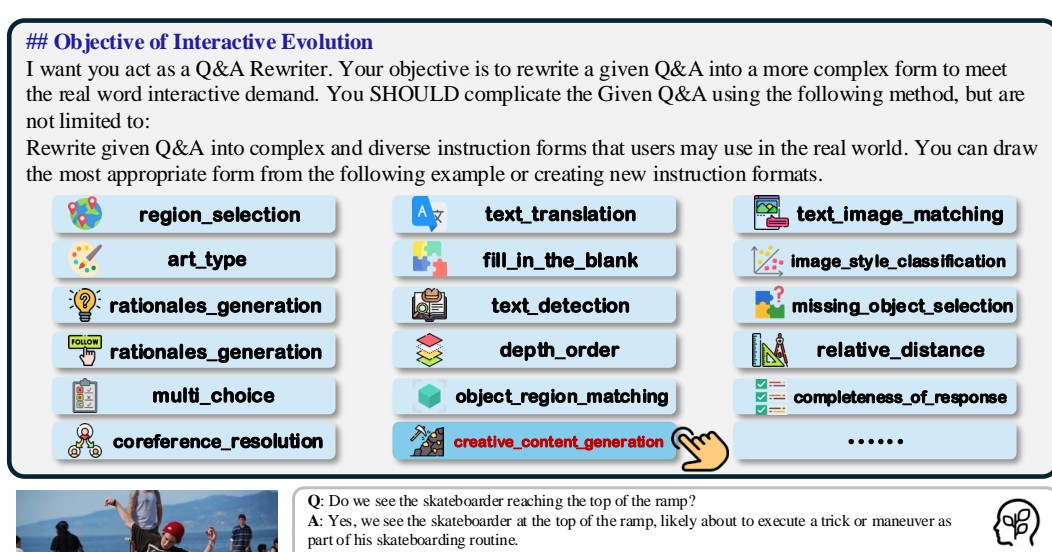

Figure 6: **Interactive evolution prompt template and example.** Interactive evolution can automatically generate various types of non-predefined instruction formats, significantly enhancing the diversity of the data. The differences are highlighted using distinct colors for better visualization.

new visual objects, uncovering usable and often overlooked visual information. The evolutionary prompt template and process are shown in Fig. 4.

---

## Instruction Elimination

Evaluate the difficulty and complexity of each rewritten samples compared to its original ones. Determine if the rewritten sample has improved in quality via "yes/no".
Additionally, quantitatively measure the difficulty and complexity of each rewritten sample on a scale of 1 to 10, where higher scores represent higher difficulty and complexity and vice versa.
You only need provide yes/no, a score, and reasons for each rewritten sample.

{"improved": "yes/no", "score": 1-10, "reason": the reason for the improvement and score},

## Evaluation Criteria

- Length: Longer Q&A pairs generally have more detail and thus are considered more complex.
- Semantic Complexity: Use of more sophisticated language or concepts.
- Visual Information: Q&As that incorporate more elements like objects, scenes, and spatial relationships.
- Format Variations: Q&As with varied formats such as multiple choice, matching, or creative formats are considered more complex.
- Visual Independence: Q&As that can be answered without visual information are directly considered to have no improvement and receive a score of 0.
Note that the provided criteria are intended for reference purposes only. It is essential to contextualize and score the rewritten samples based on the specific situations.

## In-context QA samples with different difficulties (1-10)

---

Figure 7: **Instruction elimination prompt template.** Instruction elimination is used to calculate the evolutionary gain and complexity level of the instruction data. We filter out harmful data that failed to evolve based on the evolutionary gain.

**Cognitive Reasoning Evolution.** Reasoning ability is one of the key capabilities of multi-modal large language models. However, most existing instruction data, such as LLaVA-Instruct (Liu et al., 2024b), consists of simple question-and-answer pairs that lack detailed reasoning processes, making it difficult for trained models to accomplish complex tasks requiring reasoning capabilities, such

as multi-modal agents and visual reasoning. We introduce the concept of a visual manipulation chain, abstracting four vision-centric reasoning capabilities into four visual operation functions described in text. By generating the necessary visual reasoning steps to solve problems, we define the complexity of the instruction data. During the cognitive reasoning evolution process, we evolve new instruction data by increasing the visual reasoning steps in the data to obtain more complex data. The evolutionary prompt template and process are shown in Fig. 5.

**Interactive Evolution.** Existing models generate instruction data in very few forms. For example, LLaVA-Instruct provides only dialogue-based question-answering, complex reasoning, and global description tasks. Handcrafted instruction data, such as ALLaVA (Chen et al., 2024a), are limited by annotators' experience, making it challenging to design various task forms. Models trained with such data often struggle to follow complex and diverse user-specified instructions or goals, limiting their practicality and applicability in real-world scenarios. To evolve instruction data with rich task forms and provide a good interaction experience, we design interactive evolution to generate instruction data with diverse task forms automatically. The evolutionary prompt template and process are demonstrated in Fig. 6.

**Instruction Elimination**. After each round of evolution, we score the evolved instruction data on multiple dimensions to assess the success of the evolution. We retain instruction data with evolutionary gains and discard those with failed evolution. The evolutionary elimination prompt template and process are shown in Fig. 7.

## 3 EXPERIMENTS

### 3.1 BENCHMARKS

To comprehensively evaluate the effectiveness of our evolutionary method, we select 13 benchmarks, with their sources and tested skills illustrated in Table 5. MIA (Qian et al., 2024) is an open-domain instruction-following benchmark that thoroughly tests the model's instruction-following abilities using extensive instruction data. MM-Self-Instruct (Zhang et al., 2024a) is a novel visual reasoning benchmark that focuses on the model's visual perception capabilities and performs common visual reasoning tasks encountered in daily life.

### 3.2 IMPLEMENTATION DETAILS

**Data.** During the pre-training phase, we use LLaVA-Pretrain-595K (Liu et al., 2024b) for image-text alignment training. In ablation experiment settings, we fine-tune using both seed data and evolved data separately to ensure a fair comparison and validate the benefits of **MMEvol**. In SOTA setting experiments, we fine-tune using evolved instruction data combined with other publicly available datasets sampled from Cambrain-1 (Tong et al., 2024) and compare it with other methods. Additional details on training data recipes can be found in the Appendix C.

**Model.** We follow the architecture from LLaVA-NeXT, where a multimodal large model consists of three key components: an LLM for next token prediction, a visual encoder for extracting visual features, and an image-text projector to align the visual and text modalities. We use Llama3-8B-Instruct (Touvron et al., 2023) for ablation experiments. For comparisons with other methods, we switch to our previous SOTA settings with Llama3-8B-Instruct and Qwen2-7B-Instruct (Bai et al., 2023). We adapt CLIP-ViT-L (Radford et al., 2021) for the visual encoder and use simple linear layers to bridge the image and text modalities.

**Training Strategies.** We conduct **MMEvol** training following widely used two-stage settings. Vision-Language Pre-training and Visual Instruction-tuning. The language models and ViT are separately pre-trained, while the projector is randomly initialized. To initially align the feature space between the visual and text modalities, we utilize the aligned dataset. Finally, we perform instruction tuning of the pre-trained model on visual language instruction datasets. Our experiments are conducted with 8×A100 GPUs and a global batch size of 128. We employ AdamW optimizer (Loshchilov, 2017) with learning rates $5 \times 10^{-5}$ and $2 \times 10^{-5}$ for aforementioned two stages respectively. Each stage is trained with one epoch with a 3% warmup strategy. Please refer to the Appendix C for more details.

### 3.3 QUALITATIVE ANALYSIS

We randomly sample 30K data points from the seed data and conduct qualitative analysis on the instruction data before and after evolution. As shown in Fig. 9, the evolved data is notably more complex. Specifically, each evolved instruction involves 0.68 more atomic abilities in Fig. 9a and has an average visual operation chain reasoning length of 0.86 longer compared with pre-evolution in Fig. 9b. As we can see from Fig. 9c, the average difficulty score of each evolution round increases progressively, demonstrating the effectiveness of cognitive reasoning evolution in increasing instruction data complexity.

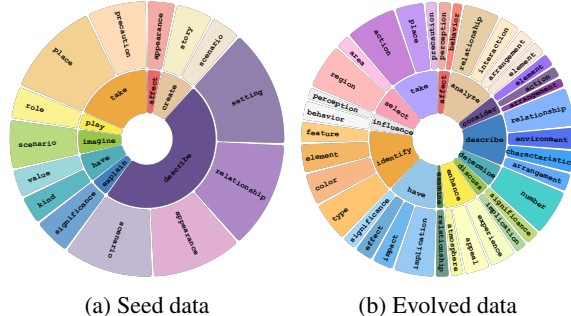

(a) Seed data     (b) Evolved data

Figure 8: The root verbs (inner circle) and their top noun objects (outer circle) of the seed data in (a) and the evolved data in (b).

We identify the verb-noun structures in the generated instructions to study the types of instructions generated and the diversity of evolved data. We use the Berkeley Neural Parser (Kitaev & Klein, 2018; Kitaev et al., 2018) to parse the instructions, extracting the verb closest to the root and its first direct noun object. Fig. 8 plots the root verbs and their direct noun objects with quantities exceeding 2K. We observe that the evolved data significantly enhances instruction diversity compared to pre-evolution, with diverse intents and textual formats in the evolved instructions. Furthermore, we conduct a long-tail distribution visualization analysis of the visual object domain in the instruction data before and after evolution to verify the effectiveness of fine-grained perceptual evolution. Fig. 10 shows that fine-grained perceptual evolution greatly improves the distribution of visual objects in the long tail, maximizing the extraction of usable visual information from images, refining the image-text alignment granularity in the instruction data, enhancing data diversity, which improves model generalization and reduces visual hallucinations.

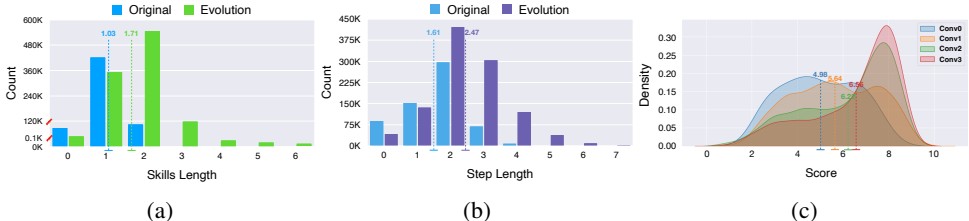

(a)     (b)     (c)

Figure 9: (a) The skills length distribution between the seed data and our evolved data; (b) The reasoning steps length distribution between the seed data and our evolved data; (c) The difficulty and complexity level distribution between the seed data and our evolved data.

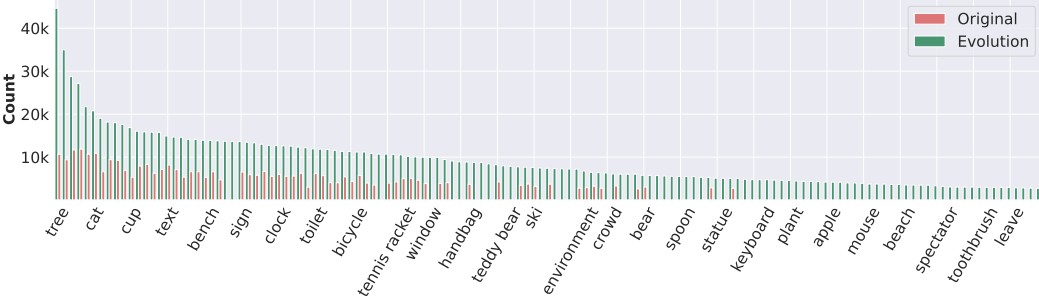

Figure 10: The long-tail distribution of 200 visual objects between seed and evolved data. **MMEvol** significantly improves the long-tail distribution of visual objects in the seed data, providing more fine-grained visual information, thereby boosting the model's generalization ability and robustness against hallucinations.

## 3.4 ABLATION STUDY

We conduct ablation studies on seven vision-language benchmarks to explore the effects of instruction evolution and elimination. As shown in Table 1, different evolution process can be orthogonally superimposed on each other to continuously enhances data diversity and complexity. leading to an average performance gain of 3.8 points across multiple vision-language benchmarks. However, the absence of instruction elimination introduces harmful data from failed evolutions, which inevitably reduces the model's resistance to hallucinations by 1.2 points on POPE (Li et al., 2023c). When both instruction evolution and instruction elimination are employed, instruction elimination filters out harmful data from failed evolutions, further improving the quality and density of evolved data and enhancing the model's performance by 0.9 points on average, particularly improving resistance to hallucinations by 1.7 points, which aligns with our qualitative analysis results in Section 3.3.

Table 1: **Ablation study on instruction evolution and instruction elimination.** The application of instruction evolution alone enhances the complexity and diversity of the data, whereas the integration of instruction elimination further refines data quality, markedly reducing the occurrence of visual hallucinations.

| FP-Evol | I-Evol | CR-Evol | I-Elim | MMStar | MathVista$^{\mathrm{M}}$ | POPE | AI2D | MME$^{\mathrm{C}}$ | MMMU$^{\mathrm{V}}$ | RWQA | AVG. |
|---------|--------|---------|--------|--------|-----------|------|------|------|-------|------|------|
| ✗ | ✗ | ✗ | ✗ | 36.5 | 25.3 | 84.8 | 53.9 | 31.5 | 32.3 | 43.5 | 44.0 |
| ✔ | ✗ | ✗ | ✗ | 37.3 (+0.8) | 25.6 (+0.3) | 85.0 (+0.2) | 54.2 (+0.3) | 33.0 (+1.5) | 32.5 (+0.2) | 46.7 (+3.2) | 44.9 (+0.9) |
| ✔ | ✔ | ✗ | ✗ | 38.2 (+1.7) | 26.2 (+0.9) | 83.8 (-1.0) | 54.5 (+0.6) | 35.6 (+4.1) | 32.9 (+0.6) | 48.9 (+5.4) | 45.8 (+1.8) |
| ✔ | ✔ | ✔ | ✗ | 38.9 (+3.4) | 27.3 (+3.0) | 83.6 (-1.2) | 54.7 (+0.8) | **40.1** (+8.6) | 34.4 (+0.9) | 54.4 (+10.9) | 47.6 (+3.8) |
| ✔ | ✔ | ✔ | ✔ | **40.3** (+3.8) | **28.6** (+3.6) | **86.5** (+1.7) | **55.2** (+1.3) | 39.9 (+8.4) | **35.3** (+3.0) | **55.3** (+11.8) | **48.7** (+4.7) |

## 3.5 BENCHMARK COMPARISON

After comprehensively validating our approach's ability to enhance the complexity and diversity of instruction data, we perform a thorough comparison with previous SOTA methods across 13 vision-language benchmarks, summarizing the results in the Table 2. Notably, we observe that supported by enhanced and refined instruction data, our MLLM significantly advances performance boundaries in almost all benchmarks, consistent with the performance improvements observed in our ablation experiments in Section 3.4. Remarkably, compared to the fully open-source SOTA model Cambrian-1 (Tong et al., 2024), our method, although using seed data sampled from training data of Cambrian-1, achieves superior results with a substantial performance increase (↑ 2.9 average points). This indicates that the quality of instruction data is more crucial than quantity.

In comparison to the open-source SOTA model MiniCPM-v2.5 (Yao et al., 2024), despite a considerable difference in training data volume, **MMEvol-8** still delivers better results, particularly showing improvements in instruction following, visual hallucinations, and visual reasoning with gains of ↑3.1 points on HallBench, ↑2.5 points on MIA, and ↑13.6 points on MMSInst respectively. This demonstrates that our method enhances the model's visual reasoning and instruction following, reduces visual hallucinations, and improves other general capabilities, consistent with our findings from ablation studies and qualitative analyses. By using our data and the leading large language model Qwen2, we can train a superior MLLM from scratch in only one day using 4×8 A100 GPUs, further validating that high-quality instruction data is more important than large-scale low-quality data.

## 4 RELATED WORK

**Multimodal Large Language Models (MLLMs).** MLLMs have rapidly advanced in recent years due to the success of Large Language Models (LLMs) and the availability of diverse image-text instruction data from the internet. LLaVA (Liu et al., 2024b) and MiniGPT-4 (Zhu et al., 2023) have demonstrated strong cross-task generalization by integrating visual encoders with large language models through simple connectors and training on instruction data. LLaVA-NeXT (Liu et al., 2024a) has significantly enhanced visual perception by employing dynamic resolution techniques. Cambrian-1 (Tong et al., 2024) has improved model robustness through visual encoder routing, though it incurs higher training costs. DEEM (Luo et al., 2024) simplifies model architecture and enhances robustness by using diffusion models to extract visual features instead of traditional visual encoders. Subsequent

Table 2: Comparison with state-of-the-art methods on 13 visual-language benchmarks. Our models consistently improve LLaVA-NeXT under a head-to-head comparison, using the same prompts and the same base LLM, showing the effectiveness of enhanced pretraining data quality. "PT" denotes pre-training data scale, "IT" denotes instruction tuning data scale and "*" denotes the baseline model trained on the seed dataset. We mark the best performance **bold** and the second-best underlined.

| Model | PT | IT | VQA$^{v2}$ | GQA | MME$^C$ | MMStar | HallBench | MathVista$^M$ | MMMU$^V$ | AI2D | POPE | MIA | BLINK | RWQA | MMSInst | AVG. |
|---|---|---|---|---|---|---|---|---|---|---|---|---|---|---|---|---|
| **Weight Open-Source** | | | | | | | | | | | | | | | | |
| Yi-VL-6B | 125M | 1M | – | – | 46.2 | 37.7 | 55.7 | 28.8 | 40.3 | 59.8 | 82.5 | 26.1 | 38.7 | 53.5 | – | 46.9 |
| DeepSeek-VL-7B | 275M | 50M | – | – | 37.1 | 40.5 | 53.9 | 36.8 | 38.3 | 65.3 | 85.6 | 61.0 | 40.9 | 49.7 | 26.7 | 48.7 |
| Qwen-VL-Chat-7B | 1.4B | 50M | 78.2 | 57.5 | 49.0 | 34.5 | 56.4 | 34.9 | 37.0 | 63.0 | 74.9 | 63.1 | 28.2 | 49.3 | – | 52.2 |
| CogVLM-Chat-17B | 1.5B | 5.1M | – | 65.2 | 37.4 | 39.9 | 55.1 | 34.7 | 37.3 | 63.3 | 88.0 | 60.0 | 41.5 | 60.3 | – | 53.0 |
| MiniCPM-V2.5-8B | 570M | 9.1M | 81.9 | 64.7 | 50.3 | 51.3 | 59.2 | 54.3 | 43.0 | 78.3 | 86.7 | 76.3 | 36.7 | 63.5 | 28.2 | 59.6 |
| InternVL2-8b | - | - | - | - | 71.8 | 61.5 | 63.9 | 58.3 | 51.2 | 83.6 | 84.2 | - | - | 64.2 | - | 67.3 |
| Qwen2-VL-7b | - | - | - | - | 64.7 | 60.7 | 68.5 | 61.4 | 53.7 | 83.0 | 85.4 | - | - | 70.1 | - | 68.4 |
| **Fully Open-Source** | | | | | | | | | | | | | | | | |
| InstructBLIP-7B | 0.6M | 0.8M | – | 49.2 | 31.8 | 32.7 | 53.6 | 24.4 | 30.6 | 40.6 | 86.1 | 38.2 | 39.7 | 36.9 | – | 42.2 |
| LLaVA-1.5-7B | 0.6M | 0.8M | 78.5 | 62.0 | 37.8 | 33.1 | 48.8 | 25.6 | 35.7 | 55.5 | 86.1 | 62.2 | 38.0 | 54.8 | 15.4 | 48.7 |
| LLaVA-1.5-13B | 0.6M | 0.8M | 80.0 | 63.3 | 34.8 | 34.3 | 45.3 | 27.7 | 37.0 | 61.1 | **88.4** | 63.6 | 40.9 | 55.3 | – | 52.6 |
| LLaVA-NeXT-8B | 0.6M | 0.8M | 81.8 | 65.2 | 44.6 | 43.9 | 52.3 | 31.5 | 41.7 | 69.9 | 87.3 | 65.1 | 43.5 | 60.1 | 25.6 | 54.8 |
| LLaVA-NeXT-13B | 0.6M | 0.8M | 82.8 | 65.4 | 37.1 | 40.4 | 51.5 | 35.1 | 35.9 | 72.2 | 87.8 | 69.2 | 41.2 | 59.1 | 30.2 | 54.5 |
| VILA-1.5-8B | 50.5M | 6.0M | 80.9 | 61.9 | 39.0 | 39.7 | 55.8 | 37.3 | 36.9 | 58.8 | 85.5 | 66.1 | 37.0 | 43.3 | 21.6 | 51.1 |
| VILA-1.5-13B | 50.5M | 6.0M | 82.8 | 64.3 | 38.5 | 44.2 | 59.2 | 42.5 | 37.9 | 69.9 | 84.2 | 61.2 | 41.5 | 53.3 | 30.6 | 54.6 |
| Cambrian-1-8B | 2.5M | 7.0M | 81.2 | 64.6 | 41.1 | 50.7 | 47.8 | 47.0 | 41.8 | 74.6 | 86.4 | 68.7 | 44.9 | **64.2** | 28.3 | 57.1 |
| Cambrian-1-13B | 2.5M | 7.0M | 82.6 | 64.3 | 44.5 | 47.1 | 58.9 | 47.4 | 40.0 | 73.6 | 86.8 | 69.8 | 43.1 | 63.0 | 25.8 | 57.5 |
| **LLaVA-NeXT*-8B** | 0.6M | 1.1M | 82.5 | 64.8 | 41.3 | 47.4 | 60.8 | 47.7 | 38.0 | 72.1 | 85.3 | 69.4 | 44.2 | 59.9 | 26.2 | 56.9 |
| **LLaVA-NeXT*-Qwen2-7B** | 0.6M | 1.1M | 82.5 | 64.9 | 44.6 | 48.9 | 61.7 | 49.3 | 41.7 | 73.3 | 86.4 | 70.2 | 44.7 | 61.0 | 30.1 | 58.4 |
| **MMEvol-8B** | 0.6M | 1.6M | **83.4** | 65.0 | 47.8 | 50.1 | 62.3 | 50.0 | 40.8 | 73.9 | 86.8 | **78.8** | 46.4 | 62.6 | 32.3 | 60.0 |
| **MMEvol-Qwen2-7B** | 0.6M | 1.6M | 83.1 | **65.5** | 55.8 | 51.6 | 64.1 | 52.4 | 45.1 | 74.7 | 87.8 | 77.6 | 47.7 | 63.9 | 41.8 | 62.4 |

work (Wang et al., 2024b; Zhou et al., 2024; Xie et al., 2024) following DEEM combine diffusion models with LLMs to further enhance generative and understanding capabilities of MLLMs. However, these models still face challenges related to the quantity and quality of data, which limit performance improvements further.

**Image-text Instruction Data Construction.** LLaVA (Liu et al., 2024b) has improved model capabilities by utilizing LLaVA-Instruct (Liu et al., 2024b), a dataset labeled by advanced LLMs. However, this approach does not fully exploit visual information and have limited instruction types. ALLaVA (Chen et al., 2024a), by manually crafting and rewriting instruction data, offers greater variety but suffers from high manual labeling costs, inefficiency, and overly simplistic problems. MMInstruct (Liu et al., 2024c) generates instruction data automatically with advanced MLLMs, but the instruction complexity and diversity are constrained by predefined formats, failing to fully exploit effective visual information. VILA$^2$ (Fang et al., 2024) has generated extensive data through instruction evolution but lacks complexity and variety, limiting its utility for other models. In contrast, we address this challenge and propose **MMEvol**, which iteratively enhances instruction diversity and complexity through instruction evolution on limited data, aiming to extract more usable visual information and endow MLLMs with more powerful capabilities.

## 5 CONCLUSIONS

In this work, we propose an image-text instruction evolution framework and explore the techniques, insights, and benefits of Evol-Instruct for enhancing the quality and quantity of image-text instruction data. We employ three distinct evolution methods to increase the complexity and diversity of instruction data based on a limited seed dataset while utilizing instruction elimination to filter out harmful data. The data evolved through three rounds of evolution is used to train a new model, demonstrating state-of-the-art (SOTA) performance across a comprehensive set of benchmarks. Future directions include exploring integrating image generation models to synthesize new images and perform dual evolution of images and texts, aiming to train even more robust foundational models.

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

APPENDIX

## A    LIMITATION

Due to resource limitation, we only performed evolution on 163K samples (approximately 12% in original data recipes) and conducted experiments with an 8B scale model. Expanding the dataset and using larger-scale models could yield even better results. We plan to explore these avenues in future work and replace the OpenAI GPT4o-mini API with open-sourced model like QWen2VL.

## B    CURATION DETAILS OF SEED DATA

LLaVA-Instruct (Liu et al., 2024b) is a dataset of image-text instructions based on the COCO (Chen et al., 2015) data source and generated using the OpenAI ChatGPT API. The image-text instruction format in this dataset primarily includes three types: dialogue-based question-answering, global descriptions, and complex reasoning. ShareGPT4V (Chen et al., 2023), on the other hand, is a dataset constructed or rewritten using the OpenAI GPT-4V API, based on image-text pairs from SAM (Kirillov et al., 2023), COCO, and other sources to introduce richer details into captions. Both LLaVA-Instruct and ShareGPT4V significantly advance the development of MLLMs (Hu et al., 2022; Li et al., 2023d; Si et al., 2024b) and are widely used. We integrate samples from these two datasets containing the same image by concatenating the corresponding instruction data lists. For samples with global descriptions but no instruction data, we use the GPT-4o-mini API to supplement the missing instruction data, similar to LLaVA-Instruct, resulting in a combined dataset of 133K samples. To ensure the diversity of the seed data, we also include additional scientific chart data. Specifically, we sample 30K entries from Cambrain-1 (Tong et al., 2024), covering various types of image-text instructions such as code generation, chart interpretation, scientific question-answering, document understanding, and mathematical reasoning, ultimately forming a seed dataset of 163K image-text instructions.

Table 3: The mixture of training recipe datasets with corresponding categories and sources. We collect these public dataset form internet.

| Category | Sources | Size | Ratio |
|---|---|---|---|
| VQA | VQAV2 (Goyal et al., 2017) | 83K | 5.1% |
| Knowledge | OKVQA (Marino et al., 2019), A-OKVQA (Schwenk et al., 2022) VG (Krishna et al., 2017), GeoQA (Gao et al., 2023) | 243K | 14.9% |
| Reasoning | GQA (Hudson & Manning, 2019) | 72K | 4.4% |
| Grounding | RefCOCO (Kazemzadeh et al., 2014) | 48K | 2.9% |
| OCR | OCR-VQA (Mishra et al., 2019), TextVQA (Singh et al., 2019) AI2D (Kembhavi et al., 2016), ChartQA (Masry et al., 2022) DocVQA (Mathew et al., 2021), DVQA (Kafle et al., 2018) Synthdog-EN (Kim et al., 2021), Datikz (Belouadi et al., 2023) TabMWP (Lu et al., 2022b), ArxivQA (Li et al., 2024) | 270K | 16.5% |
| Instruct | **MMEvol**, ALLaVA (Chen et al., 2024a) | 650K | 39.8% |
| Language | ShareGPT, WizardLM (Xu et al., 2023) | 183K | 11.2% |
| Science/Code | Design2Code (Si et al., 2024a), MathVision (Wang et al., 2024a) Geo170k (Gao et al., 2023), ScienceQA (Lu et al., 2022a) Websight (Li et al., 2023a), Cambrain-Data-Engine (Tong et al., 2024) | 85K | 5.2% |

## C    IMPLEMENTATION DETAILS

After three rounds of evolution and filtering, we obtain 447K high-quality image-text instruction data with diversity and complexity. This data, combined with the ALLaVA instruction dataset, forms the 600K instruction data segment of the training recipe. To ensure a fair comparison with other methods, we combine the instruction data with other commonly used image-text data into the final training recipe, as shown in the Table 3. Notably, we find that the DataEngine (Tong et al., 2024) data contains many harmful mismatched image-text pairs. We use OpenAI GPT-4o API to filter out harmful data

Table 4: The detailed training setup for **MMEvol** and the hyper-parameters across the training stages.

| Hyperparameter | Ablation Stage 1 | Ablation Stage 2 | SOTA Stage 1 | SOTA Stage 2 |
|---|---|---|---|---|
| language model | LLaMA 3 8b | LLaMA 3 8b | LLaMA 3 8b Qwen 2 7b | LLaMA 3 8b Qwen 2 7b |
| global batch size | 128 | 128 | 128 | 128 |
| batch size | 4 | 4 | 4 | 4 |
| learning rate | 1e-3 | 5e-5 | 1e-3 | 5e-5 |
| lr schedule | cosine | cosine | cosine | cosine |
| lr warmup ratio | 0.03 | 0.03 | 0.03 | 0.03 |
| weight decay | 0 | 0 | 0 | 0 |
| epoch | 1 | 1 | 1 | 1 |
| optimizer | AdamW | AdamW | AdamW | AdamW |
| cost | 4h | 0.1h | 4h | 20h |
| dataset | LLaVA Pretrain | Seed-30K/Evol-30k | LLaVA Pretrain | Dataset Mixture |

and obtain 20K effective image-text instruction data. More details about training settings can be found in Table 4

## D  ADDITIONAL VISUALIZATION RESULTS

We sample a specimen from SEED-163K and display its evolution process in Fig. 11. In round 1, we perform fine-grained perceptual evolution, leading to instruction data with more precise details, including actions and attributes. In round 2, interaction evolution shifts instruction forms from general question answering to creative poetry generation, increasing the diversity of instruction formats. In round 3, cognitive reasoning evolution adds reasoning steps to the answers in the instruction data, enhancing its complexity. Through multiple rounds of instruction evolution, we improve the diversity and complexity of the seed data.

We plot the performance of the model at every 1k step across 9 evaluation datasets in Fig. 12 to observe the learning trends during training. We can observe that the model learns OCR-related capabilities and mathematical reasoning abilities relatively smoothly, while general perception and cognitive skills exhibit more challenges. This may stem from conflicts arising from multi-source training tasks. A phased learning approach based on the difficulty of different tasks could be adopted to achieve better performance. We also present additional visualization results to demonstrate the capabilities of our model. As shown in Fig. 13, our model trained on this data exhibits strong visual reasoning, instruction following, and fine-grained perception capabilities. Additionally, it identifies nuances in meme content, validating the effectiveness and efficiency of **MMEvol**.

## E  COMPLETE EVOLUTION PROMPT TEMPLATE

Due to the space limitations in the main text, we simplify the instruction evolution prompt template. We provide the complete detailed evolution templates as follows: the complete prefix-prompt template is shown in Fig. 16, the fine-grained perception evolution prompt template is in Fig. 17, the cognitive reasoning evolution prompt template is in Fig. 18, the interaction evolution prompt template is in Fig. 19, and the instruction elimination prompt template is in Fig. 20.

Table 5: Benchmarks for evaluation with their sources and tested skills. The names are abbreviated due to space limitations. $VQA^{V2}$; GQA; $VQA^T$: TextVQA; $MME^C$: MME-Cognition; $MathVista^M$: MathVista-MINI; MMMU; AI2D; POPE; HallusionBench: HallBench; MIA; BLINK; RWQA: RealWorldQA; MMSInst: MM-Self-Instruct.

| Skills | Sources | Skills | Sources |
|---|---|---|---|
| VQA | $VQA^{v2}$ (Goyal et al., 2017) | General Knowledge | $MME^C$ (Fu et al., 2023) |
| Knowledge Leakage | MMStar (Chen et al., 2024b) | | MMMU (Yue et al., 2024) |
| Math Reasoning | $MathVista^M$ (Lu et al., 2023) | Hallucination | POPE (Li et al., 2023c) |
| OCR Related | AI2D (Kembhavi et al., 2016) | | HallBench (Guan et al., 2023) |
| Instruction Following | MIA (Qian et al., 2024) | Visual Reasoning | GQA (Hudson & Manning, 2019) |
| Visual Perception | BLINK (Fu et al., 2024), RWQA | | MMSInst (Zhang et al., 2024a) |

| Seed 🌱 |
| --- |
| **Question-seed**: How many players are wearing orange jerseys?
**Answer-seed**: There are three players wearing orange jerseys. |
| **Base Prompt** |
| **Question**: How many individuals in the image are dressed in orange jerseys??
**Answer:** There are three individuals dressed in orange jerseys. |
| **Fine-grained Perceptual Evolution** 🌿 |
| **Question-R1:** Observe the player wearing the **orange jersey with the number 8**. **How is their body positioned and what action are they engaged in**?
**Answer-R1:** The player in the orange jersey with the number 8 is in **mid-air, poised to take a shot, with both hands holding the ball**. |
| **Interactive Evolution** 🌳 |
| **Question-R2:** Given the basketball game depicted in image, **create a poem that reflects the atmosphere of the moment captured**.
**Answer-R2:** On the vibrant court where spirits soar,\nThe game unfolds with passion's roar,\nPlayers leap and chase the score,\nIn pursuit of dreams and more.\nThe scoreboard's light flickers bright,\nEach play ignites the crowd's delight,\nIn this arena of youthful fight,\nWe see the glory of the night. |
| **Cognitive Reasoning Evolution** 🌳 |
| **Question-R3:** Based on the information in the image, **which color team is likely to win?**
**Answer-R3:** **To determine** which team might win, **first**, we look at the scoreboard in the top-left corner of the image, which shows a score of 71:55. The team names on the scoreboard are not clearly visible. **Next**, we observe a blue player in the center of the image whose jersey displays the text 'MORABANC'. Given that the score of 71 corresponds to a team name that **can be identified as** 'MORABANC' from the jersey, and knowing the team's color is blue, **we infer that the blue team** (MORABANC) is likely to have an advantage in the current period. |

Figure 11: **MMEvol** continuously enhances instruction data complexity and diversity over evol-instruct. The sample is from SEED-163K. We mark fine-grained visual information in red, new instructions form in green, and longer reasoning steps in blue. Additionally, we also provide the rewritten samples via a base prompt for comparison.

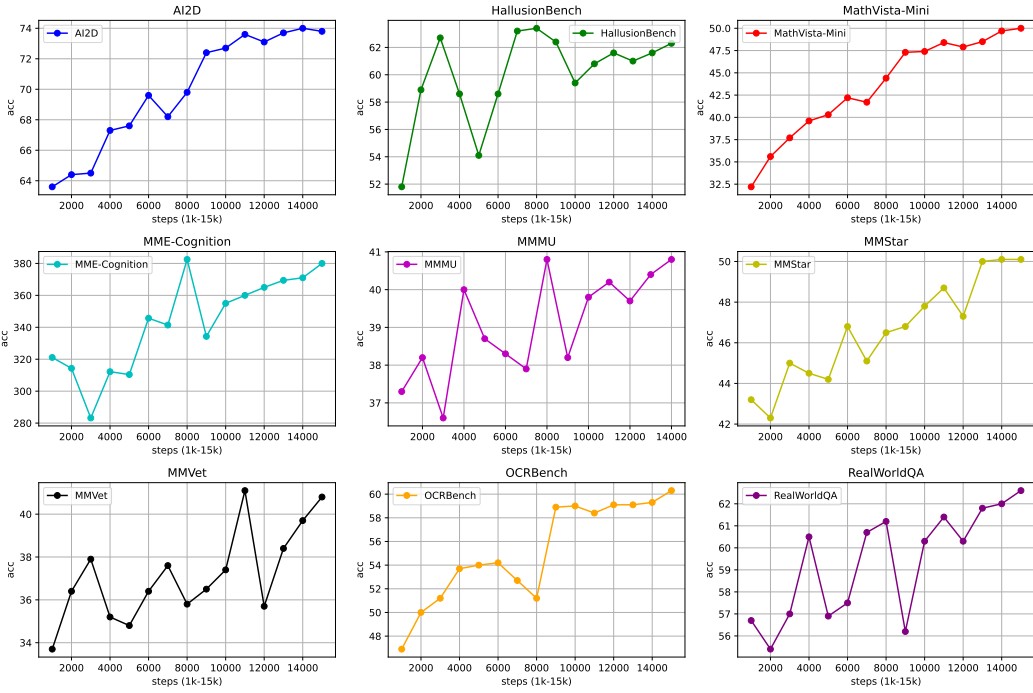

Figure 12: **Illustration of Performance Trends Across Various Benchmarks.** We plot the performance of the model at every 1k step across 9 evaluation datasets to observe the learning trends during training.

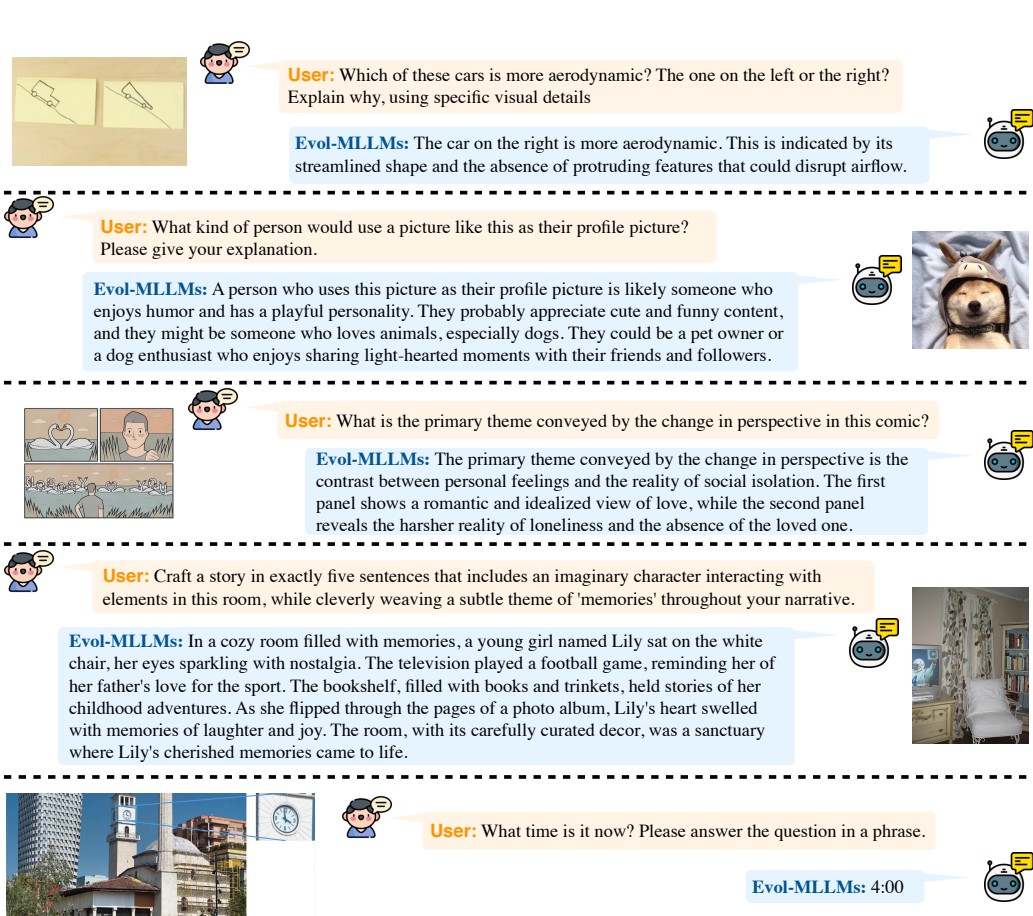

Figure 13: **Examples of image-text dialogue with our MMEvol-8B Model.** Training our model on the enhanced data not only enables it to handle handwritten mathematical reasoning, complex instruction following, and the understanding of emotional nuances, but also allows it to perceive fine-grained information within images, demonstrating the effectiveness of our approach.

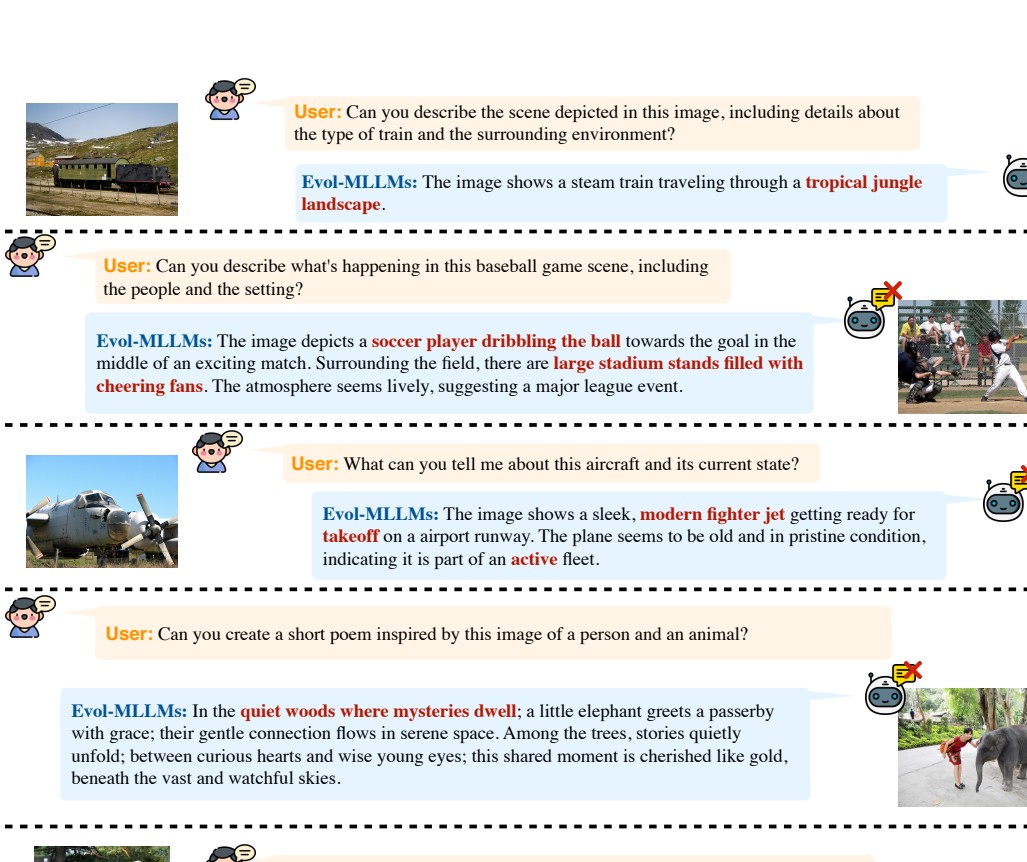

Figure 14: **Some failed cases of image-text dialogue with our MMEvol-8B Model.** Our model generalizes to multiple visual reasoning scenes when training on the instruction data via our evolution strategy. However, there can still be errors in some scenarios, *e.g.*, hallucination.

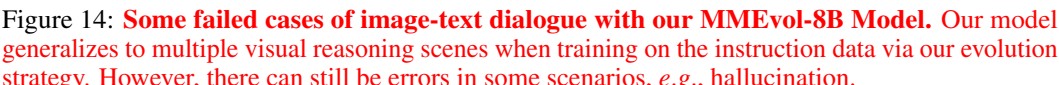

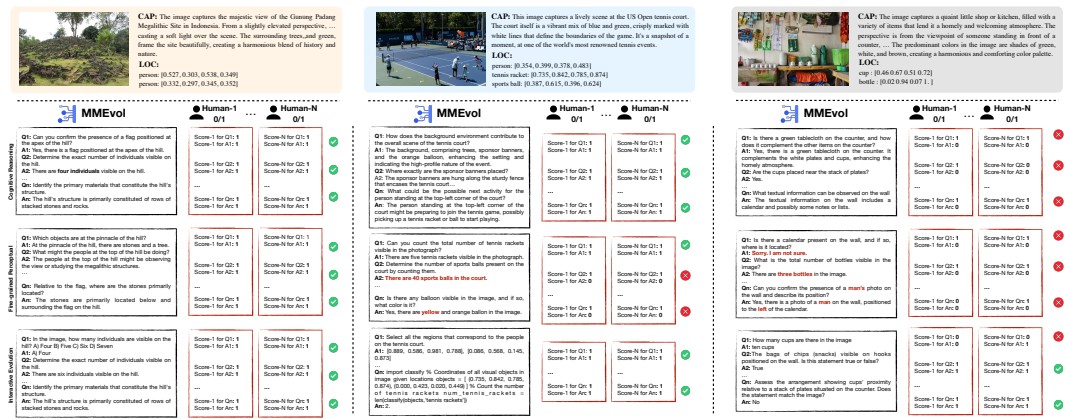

Figure 15: **Schematic diagram of the manual filtering process.** We hired N=5 experts to score (0/1) each question and answer. In the event that any question or answer receives a score of 0, the entire QA pair will be deemed invalid and discarded.

## Context Type I: Caption

The image shows a modern living room with natural light streaming through a large window... A black couch against a gray wall, …, a glass coffee table that holds a white vase and a plant... The table rests on a beige rug, contrasting with the hardwood floor, adding warmth. The design suggests a comfortable and stylish living area.

## Context Type II: Visual Object Locations

window : [0.2 0.23 0.57 0.4], couch : [0.17 0.43 0.83 0.79], vase : [0.5 0.51 0.58 0.72] …

--------------------------------------------------------------------------------

## Vision-Centered Multimodal Atomic Propositions & Permitted Vision-Centric Manipulations

1. Grounding Ability: Given a description of a visual object, output the coordinates of the visual object in the image and a natural language explanation.
2. Referencing Ability: Given the coordinates of a visual object, output the corresponding visual object description.
3. Calculating Ability: Ability to calculate the number, size, and other information of visual objects in the image and obtain the corresponding numbers.
4. OCR Ability: Recognize and generate textual representations of structured data in the image, such as numbers, text, codes, tables, etc.
5. Existence Ability: Given a description of a visual object, determine whether it exists in the image.

### Permitted Vision-Centric Manipulations and Their Usage Descriptions

- Grounding_i(tgt)->bbx_i: The i-th grounding manipulation, that locates the object(s) specified by the target noun phrase `tgt` in the current image, and returns the resulting bounding box(es) as `bbx_i` where each box is represented by the top-left and bottom-right coordinates.
- Referring_i(bbx)->tgt_i: The i-th referencing manipulation, used to identify small and subtle objects in the image; it locates the current image using the box `bbx` defined by the top-left and bottom-right coordinates, zooms in the area by two times, and returns the resulting `tgt_i`.
- Calculate(tgt)->res_i: The i-th calculate manipulation, that calculates the formula specified by the target `tgt` in the current image, and returns the calculation result `res_i`.
- OCR_i(tgt)->txt_i: The i-th OCR manipulation, that recognizes the natural texts written on the target `tgt`, and returns the recognized texts `txt_i`.

## Language-Centered Multimodal Atomic Propositions & Permitted Vision-Centric Manipulations

1. Relationship Description Ability: Understand and recognize relationships between different visual objects in the image, such as temporal, spatial, logical, etc.
2. Context Understanding Ability: Recognize and interpret complex scenes or situations in the image, such as asking about ongoing events, implied stories, unusual meaning, etc.
3. Behavior Prediction Ability: Predict possible subsequent actions based on the image content.
4. Knowledge Integration Ability: Integrate visual objects in the image with additional world knowledge, such as asking about background knowledge related to the objects.

## Objective

…

--------------------------------------------------------------------------------

## Given Q&A

```
{
    "objects" : [window, couch, vase, …]
    "skills": ["Grounding Ability", "Context Understanding Ability", …]
    "format" : "Complex reasoning",
    "question": "Which is closer to the window, couch or vase?"
    "steps": [{"manipulation": "grounding_1(`window`)-> bbx_1"description": "Locate the window in the room and
```
return its bounding box as `bbx_1`".}, {"manipulation": "grounding_2(`couch`)-> bbx_2", "description": "Locate the couch in the room and return its bounding box as ` bbx_2`}, {"manipulation": "grounding_3(`vase`)-> bbx_3", "description": "Locate the vase in the room and return its bounding box as ` bbx_1`". ],
"answer": First, based on the detected bounding boxes of the window [0.2, 0.23, 0.57, 0.4] and the couch [0.17, 0.43, 0.83, 0.79], it can be inferred that the couch is positioned below the window. Further, according to the location of the vase [0.5 0.51 0.58 0.72], it is located near the center of the room. Therefore, the couch is closer to the window.
}

## Rewritten Q&A

Figure 16: **Complete prefix-prompt template of MMEvol.**

## Objective of Fine-grained Perceptual Evolution

I want you act as a Q&A Creator. Your objective is to draw inspiration from the given Q&A to create a brand new created Q&A. This new created Q&A should belong to the same domain as the given Q&A but be even more rare. The difficulty level of the created Q&A should be similar to that of the given Q&A. Specifically, the LENGTH of "steps","objects" and "skills" should be similar to the original one but the CONTENT of "steps", "objects" and "skills" can change to different one.

Prioritize questions with definite answers. If a question can be resolved with only a few solving steps, it can be reformulated to explicitly request additional solving steps. It is essential to avoid making the #Rewritten Q&A overly verbose.

## Constraints

- Achieve solving steps and answers related to the questions.
- Ensure all generated data is consistent with the image content.
- Double-check provided descriptions against the image content.
- Do not generate new location coordinates; use the given coordinates.
- Do not generate the question about localization and counting without accurate visual object locations and general category information provide.

## Example
...

## Given Q&A
...

## Rewritten Q&A
...

Figure 17: **Complete fine-grained perceptual evolution prompt template.**

## Objective of Cognitive Reasoning Evolution

I want you act as a Q&A Rewriter. Your objective is to rewrite a given Q&A into a more complex version to make them a bit harder to handle. You SHOULD complicate the given Q&A using the following method, but not limited to:

In the rewritten problem, include 1-2 new visual object categories and multimodal atomic propositions, while avoiding making the problem unnecessarily lengthy. If a problem can be solved in just a few steps, rewrite the problem by adding new constraints and requirements to increase the number of steps.

## Constraints

- Achieve solving steps and answers related to the questions.
- Ensure all generated data is consistent with the image content.
- Double-check provided descriptions against the image content.
- Do not generate new location coordinates; use the given coordinates.
- Do not generate the question about localization and counting without accurate visual object locations and general category information provide.

## Example
...

## Given Q&A
...

## Rewritten Q&A
...

Figure 18: **Complete cognitive reasoning evolution prompt template.**

## Objective of Interactive Evolution

I want you act as a Q&A Rewriter. Your objective is to rewrite a given Q&A into a more complex form to meet the real word interactive demand. You SHOULD complicate the Given Q&A using the following method, but are not limited to:

Rewrite given Q&A into complex and diverse instruction forms that users may use in the real world. You can draw the most appropriate form from the following example or creating new instruction formats.

Prioritize questions with definite answers.

If a question can be resolved with only a few solving steps, it can be reformulated to explicitly request additional solving steps. It is essential to avoid making the #Rewritten Q&A# overly verbose.

## Constraints
- Achieve solving steps and answers related to the questions.
- Ensure all generated data is consistent with the image content.
- Double-check provided descriptions against the image content.
- Do not generate new location coordinates; use the given coordinates.
- Do not generate the question about localization and counting if there are no accurate visual object locations and general category information provide.
- Ensure the image position in the generated Q&A&S is consistent with the given Q&A&S, and that they all belong to the same sub-image.

## Example
...

## Given Q&A
...

## Rewritten Q&A
...

Figure 19: **Complete interactive evolution prompt template.**

## Instruction Elimination

Evaluate the difficulty and complexity of each rewritten samples compared to its original ones. Determine if the rewritten sample has improved in quality via "yes/no".

Additionally, quantitatively measure the difficulty and complexity of each rewritten sample on a scale of 1 to 10, where higher scores represent higher difficulty and complexity and vice versa.

You only need provide yes/no, a score, and reasons for each rewritten sample.

{"improved": "yes/no", "score": 1-10, "reason": the reason for the improvement and score},

## Evaluation Criteria
- Length: Longer Q&As generally have more detail and thus are considered more complex.
- Semantic Complexity: Use of more sophisticated language or concepts.
- Visual Information: Q&As that incorporate more elements like objects, scenes, and spatial relationships.
- Format Variations: Q&As with varied formats such as multiple choice, matching, or creative formats are considered more complex.
- Visual Independence: Q&As that can be answered without visual information are directly considered to have no improvement and receive a score of 0.
Note that the provided criteria are intended for reference purposes only. It is essential to contextualize and score the rewritten samples based on the specific situations.

## In-context QA samples with different difficulties (1-10)
...

Figure 20: **Complete instruction elimination prompt template.**

## Cognitive Reasoning Evolution

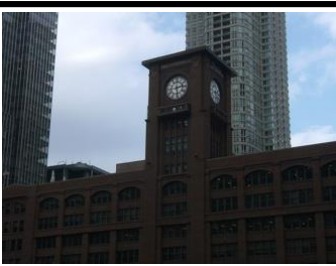

### conversations_v0

**Q:** What is the name displayed on the clock tower?
**A:** The name 'BRITANNIA' is displayed on the clock tower.

**Q:** Where is the clock located on the building?
**A:** The clock is located on the upper part of the clock tower, with coordinates approximately [0.488, 0.259, 0.553, 0.348].

**Q:** How does the sky contribute to the overall atmosphere of the image?
**A:** The light blue sky dotted with fluffy white clouds contributes to a bright and harmonious atmosphere in the image. It complements the blend of old and new architectural styles, unifying the scene under a shared expanse.

### conversations_v1

**Q:** What name is prominently displayed on the clock tower, and how does it relate to the architectural features of the clock and building? Let's think step by step.
**A:** The name 'BRITANNIA' is displayed on the clock tower. This name contributes to the identity of the structure, reflecting its historical significance in the urban landscape. The brick building, characterized by its warm brown hue and arched windows, adds a sense of nostalgia and tradition that contrasts with the modern skyscrapers in the background. Together, these elements illustrate a conversation between past and present in the architectural aspects of the scene.

**Q:** Given the positioning of the clock on the building, how does this contribute to the overall function and visual prominence of the tower in the urban scene? Let's think step by step.
**A:** The clock is positioned at the upper part of the clock tower, with bounding coordinates approximately [0.488, 0.259, 0.553, 0.348]. This elevated placement enhances its visibility, allowing it to serve as a key point of orientation within the city. As a prominent feature, the clock not only fulfills its practical function of telling time but also acts as a striking landmark, drawing attention within the contrasting backdrop of the sleek modern skyscrapers nearby.

**Q:** In what ways does the sky contribute to the overall aesthetic and functional significance of the clock tower within this urban scene? Let's think step by step.
**A:** The light blue sky adorned with fluffy white clouds infuses the urban scene with brightness and tranquility. This serene backdrop not only enhances the aesthetic appeal of the clock tower but also serves to focus attention on it as a significant landmark within the skyline. The harmonious atmosphere created by the sky juxtaposes beautifully with the architectural elements of the buildings, reinforcing the clock tower's role as a central feature in the city's identity.

Figure 21: **Data case of cognitive reasoning evolution.**

## Fine-grained Perceptual Evolution

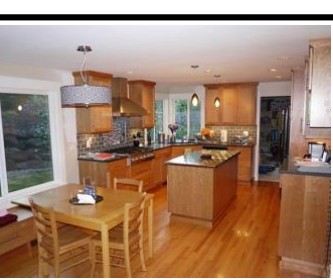

### conversations_v₀

**Q:** What material is prominently featured in the kitchen?
**A:** Wood is the prominently featured material in the kitchen, as seen in the table, chairs, cabinets, and floors.

**Q:** What is the condition of the kitchen in the image?
**A:** The kitchen appears to be immaculate, clean, and well-maintained.

**Q:** What kind of countertops does the kitchen have?
**A:** The kitchen has dark countertops that contrast with the wood elements.

**Q:** Is there an island or a small table in the kitchen?
**A:** There is an island and a small table present in the home kitchen, providing additional workspace and seating options.

**Q:** What type of flooring material is used in this kitchen?
**A:** The flooring material used in this kitchen is beautiful hardwood floors.

### conversations_v₁

**Q:** What types of appliances are integrated into the kitchen?
**A:** The kitchen is equipped with modern appliances including a gas range, refrigerator, dishwasher, microwave, and coffee maker.

**Q:** What type of lighting fixture is predominately used in the kitchen?
**A:** The kitchen features pendant lights above the island, providing style and bright illumination.

**Q:** Is there any decorative shelf visible in the kitchen?
**A:** Yes, there is a wooden shelf adorned with a plant and vase, adding a touch of nature to the kitchen.

**Q:** What type of view does the kitchen window offer?
**A:** The kitchen window offers a picturesque view of a rock wall and lush greenery outside.

**Q:** What does the dining setup in the kitchen suggest about its use?
**A:** The dining setup with a table, four chairs, and a bench suggests that the kitchen can comfortably accommodate family meals and gatherings.

Figure 22: **Data case of fine-grained perceptual evolution.**

**Interactive Evolution**

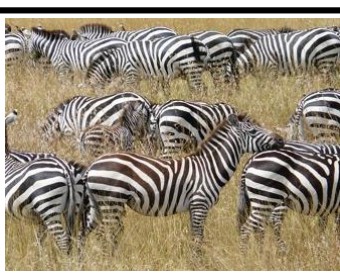

## conversations_v₀

**Q:** How many zebras are visible in the image?
**A:** There are twelve zebras visible in the picture. Their bounding box coordinates are as follows: [0.216, 0.366, 0.834, 0.946], [0.681, 0.517, 1.000, 0.937], [0.105, 0.293, 0.451, 0.578], [0.440, 0.317, 0.683, 0.551], [0.647, 0.038, 0.999, 0.337], [0.657, 0.032, 0.877, 0.156], [0.519, 0.048, 0.683, 0.321], [0.236, 0.069, 0.614, 0.321], [0.048, 0.022, 0.365, 0.235], [0.229, 0.418, 0.391, 0.557], [0.000, 0.358, 0.234, 0.950], and [0.841, 0.276, 1.000, 0.494].

**Q:** Are there any zebras grazing on grass in the image?
**A:** Yes, there are zebras grazing on grass in the image, as evidenced by some of the zebras lowering their heads to the grass.

**Q:** How many zebras are in the foreground?
**A:** There are three zebras in the foreground, represented by the larger and more prominently located zebras.

**Q:** What is the relationship between the zebras and their habitat in the image?
**A:** The zebras are seen grazing and standing alert in their natural habitat, which is a dry grass field. This type of habitat provides food and a space where they can stay in groups for safety.

## conversations_v₁

**Q:** Select the accurate number of zebras visible in the image from the options: A) 10 B) 12 C) 15 D) 20.
**A:** B

**Q:** Between the zebras located in the foreground and background, which group has a higher count?
**A:** There are three zebras in the foreground, represented by the larger and more prominently located zebras.

**Q:** Provide two rationales for how the zebras interact with their habitat depicted in the image.
**A:** 1. The zebras are seen grazing on the dry grass, utilizing it as a food source. 2. Their presence in the grassland allows them to maintain social interactions and observe their surroundings for predators.

Figure 23: **Data case of interactive evolution.**

