# OpenReview forum: "MMEvol: Empowering Multimodal Large Language Models with Evol-Instruct"
_ICLR.cc/2025/Conference — Submitted to ICLR 2025_

### Official Review · Reviewer_u5c1 · 2024-10-17

**Soundness:** 3
**Presentation:** 3
**Contribution:** 2
**Rating:** 6
**Confidence:** 4

**Summary:**

The paper addresses the challenge of enhancing the quality and diversity of training data for Multimodal Large Language Models (MLLMs). Traditional model-driven approaches face diminishing returns, while existing data-driven methods are limited by the complexity and variety of available data. To overcome this, the authors propose MMEvol, a framework that iteratively refines image-text instruction data through fine-grained perceptual evolution, cognitive reasoning evolution, and interactive evolution. This process generates a richer and more diverse dataset, improving the models' visual understanding and reasoning capabilities. The approach leads to a significant performance boost across multiple vision-language benchmarks, achieving state-of-the-art results in several tasks with less data compared to existing methods. This paper contributes by advancing the capability of MLLMs through an innovative data evolution method that emphasizes the quality of instructions over sheer data volume.

**Strengths:**

Data Evolution Framework: The MMEvol framework effectively enhances the diversity and complexity of image-text instruction data through iterative evolution methods, such as fine-grained perceptual, cognitive reasoning, and interactive evolution. This approach significantly improves the quality of data used for training MLLMs, addressing a critical bottleneck in existing data-driven methods.

Empirical Improvement: The proposed method achieves an average accuracy improvement of 3.1 percentage points over baseline models and reaches state-of-the-art performance in nine vision-language tasks. This demonstrates the efficacy of MMEvol in enhancing model capabilities with less training data compared to other approaches.

Data Balancing and Quality Analysis: The generated instructions are more compositional, longer in reasoning chains, and more balanced between objects. The statistics of the instructions suggest improved quality of instructions.

**Weaknesses:**

Evaluation Limitation: While the paper claims that the instruction generation method is a key contribution, it lacks a direct comparison with other multimodal instruction generation techniques, such as MIMIC-IT or LLaVA-NEXT. To strengthen this evaluation, I suggest that the authors generate instructions using MIMIC-IT and LLaVA-NEXT methods on the same seed data and compare the quality, diversity, and complexity of the resulting instructions with those generated by MMEvol. This would help demonstrate how MMEvol performs relative to similar methods in the field, addressing the largest weakness of the paper.

Absence of Failure Case Study: The paper does not sufficiently explore the limitations or failure scenarios of MMEvol. I recommend that the authors provide concrete examples of cases where MMEvol might fail or produce suboptimal results. Additionally, conducting an analysis of how performance changes with increasing rounds of evolution could help identify potential saturation points, offering insights into when the method might negatively impact model performance.

Technical Contribution: The paper's technical contribution feels incremental compared to previous works like MIMIC-IT[1] or MM-Instruct[2], which also iteratively refine instructions from image metadata. To clarify the novelty of MMEvol, it would be helpful for the authors to explicitly outline which aspects of their method are similar to these prior works and what specific contributions MMEvol makes beyond them. This will help position the paper relative to existing methods and highlight any unique advancements.

[1] Li et al., MIMIC-IT: Multi-Modal In-Context Instruction Tuning, 2023.

[2] Liu et al., MM-Instruct: Generated Visual Instructions for Large Multimodal Model Alignment, 2024.

**Questions:**

- Fig 12: why does negative scaling happen at step 6k-7k? Is it related to certain instruction augmentation techniques?
- Table 1: What are the number of instructions for each row?
- Table 2: What's Qwen2-7B baseline performance?
- The quality, and maybe quantity, of the seed instruction set may affect the quality of the generated instructions. Please provide a study on this to evaluate the robustness of the proposed method.

---

> ### Author Response · Authors · 2024-11-21
>
> Thanks for the valuable and encouraging comments! Our point-by-point responses to the reviewer's mentioned concerns are provided as follows.
>
> > **W 1**: Evaluation Limitation
>
> **Response:**
>
> Thank you for your valuable suggestion. In order to draw comparisons with MIMIC-IT, we constructed a dataset of 3,000 evolutionary data points using MIMIC-IT and GPT4o-mini based on a seed set of 1,000 data points. We conducted comparative experiments with MMEvol, with the results presented in the table below. Under stringent conditions of fairness (seed data, evolutionary API, model architecture), MMEvol achieved an average lead of 6.4 points, demonstrating particularly significant superiority on the RealWorldQA task. This highlights the effectiveness of MMEvol.
>
> | Seed          | MMStar | MathVista$^M$ | MME$^C$ | AI2D | HallBench | MMMU$^V$ | RWQA | AVG. |
> | ------------- | ------ | ------------- | ------- | ---- | --------- | -------- | ---- | ---- |
> | MIMIC-IT (3K) | 32.1   | 24.3          | 26.4    | 47.6 | 41.9      | 31.5     | 34.5 | 34.1 |
> | MMEvol (3K)   | 37.9   | 26.1          | 31.3    | 55.1 | 43.8      | 35.8     | 53.2 | 40.5 |
>
> > **W 2**: Absence of Failure Case Study
>
> **Response:**
>
> Thank you very much for your detailed review. We have included the missing Failure Case Study in Figure 14 and highlighted it in red.
>
> > **W 2**: Technical Contribution
>
> **Response:**
>
> We further elucidate the differences between MMInstruct, MIMIC-IT, and MMEvol from the perspectives of complexity and diversity.
>
> 1. MMEvol is characterized by its elegance and simplicity, enabling the completion of an arbitrary number of task expansions using a unified prompt and straightforward image data. In contrast, MIMIC-IT and MMInstruct employ more complex pipelines and require a limited set of pre-defined tasks. For instance, MIMIC-IT necessitates both image and video data to accomplish six specific tasks. MMEvol transcends the limitations of pre-set tasks by generating new tasks, thus eliminating the need for intricate manual multi-task designs, making it more efficient and effective, with a higher degree of diversity.
>
> 2. MMEvol continuously iterates through comparison and evolution based on existing questions and answers via an instruction-driven evolution approach, generating more complex tasks. Conversely, MIMIC-IT and MMInstruct create new data by repeatedly inputting new images into pre-defined tasks through rewriting. The former has a clear objective of increasing complexity, while the latter merely reprocesses input data within a limited task framework. This results in MMEvol achieving superior data complexity and quality with only 480K evolution data, while MIMIC-IT requires 2.8M.
>
> > **Q 1**: why does negative scaling happen at step 6k-7k? Is it related to certain instruction augmentation techniques?
>
> **Response:**
>
> The issue of multi-task training conflicts exists within the framework of multimodal instruction fine-tuning [1]. The various capabilities of multimodal models do not enhance comprehensively without conflict. It is reasonable that negative scaling occurs between steps 6k and 7k. We did not employ any instruction enhancement techniques; rather, we simply present the results of our experiments.
>
> [1] Chen Wei, et al."LLaVA-MoLE: Sparse Mixture of LoRA Experts for Mitigating Data Conflicts in Instruction Finetuning MLLMs"  *arXiv preprint arXiv:2401.16160* (2024).
>
> > **Q 2**: Table 1: What are the number of instructions for each row?
>
> **Response:**
>
> For each row, we employ 6K instructions for the ablation study. Here, we clarify that we utilized an equivalent amount of 6K data for each row.
>
> > **Q 3**: Table 2: What's Qwen2-7B baseline performance?
>
> **Response:**
>
> Thank you for your invaluable suggestions. We have added the Qwen2-7B baseline trained on the seed data (highlighted in red) to Table 2 of the paper.

---

> > ### Author Response · Authors · 2024-11-21
> >
> > > **Q 3**: Please provide a study on this to evaluate the robustness of the proposed method.
> >
> > **Response:**
> >
> > We selected 1K data points with lower scores (<5) as low-quality seed data for three rounds of instruction evolution, resulting in 3K evolved data points. We conducted a comparative experiment between these evolved data and an equal quantity of 3K data points evolved from randomly selected seed data. The results are presented in the table below and defualt setting is random initialization.
> >
> > | Seed                           | MMStar | MathVista$^M$ | MME$^C$ | AI2D | HallBench | MMMU$^V$ | RWQA | AVG. |
> > | ------------------------------ | ------ | ------------- | ------- | ---- | --------- | -------- | ---- | ---- |
> > | Random-Initialization (3K)     | 37.9   | 26.1          | 31.3    | 55.1 | 43.8      | 35.8     | 53.2 | 40.5 |
> > | Low-Score-Initialization  (3K) | 36.5   | 25.4          | 29.9    | 54.8 | 43.1      | 35.0     | 52.6 | 39.7 |
> >
> > The quality of evolutionary data is influenced by the initial instructions, although the impact is relatively minor. Nevertheless, high-quality instructional data can still be generated through multiple iterations, demonstrating the robustness of our method.
> >
> >
> >
> > Thank you again for your insightful comments.  If you have other comments, we are happy to address them to polish this work. We look forward to contributing to the development of both the Multi-Modal research and the open-source community.

---

> ### Author Response · Authors · 2024-11-22
>
> Dear Reviewer u5c1:
>
> Thank you for your extensive review and constructive comments on our manuscript. We have earnestly worked on resolving all issues highlighted and have provided comprehensive responses to your queries. As the review deadline is imminent, we humbly request your reevaluation of our revised submission. Please feel free to reach out if any further clarification is required from our side.
>
> Your support is highly valued, and we thank you in advance for your consideration.
>
> With gratitude,
>
> Authors

---

> ### Author Response · Authors · 2024-11-24
>
> Dear Reviewer u5c1,
>
> I hope this message finds you well. We have carefully considered your feedback and have made significant improvements to the manuscript. We truly value your insights, and your expertise has greatly contributed to enhancing the quality of our work. Could you please let us know if the revisions meet your expectations? We are eager to address any further queries you might have.
>
> Thank you for your invaluable support and consideration.
>
> Warm regards,
>
> Authors

---

> ### Author Response · Authors · 2024-11-25
>
> Dear Reviewer u5c1,
>
> We genuinely value the time and effort you have dedicated to reviewing our manuscript. We have responded comprehensively to your comments and made the necessary adjustments. As the deadline for discussion nears, we kindly ask if you could review our updated paper. We are eager to address any additional questions that may arise.
>
> Thank you very much for your continued support and assistance.
>
> Sincerely,
>
> Authors

---

> > ### Comment · Reviewer_u5c1 · 2024-11-25
> > **Thank you for your response**
> >
> > Thank you for your response. You have addressed most of my concerns so I'm glad to raise the rating to 6.

---

> > > ### Author Response · Authors · 2024-11-26
> > >
> > > Dear Reviewer u5c1:
> > >
> > > Thank you for your reply. Your comments are very insightful and valuable for us yo polish this paper.  If you have any additional questions, please do not hesitate to let us know. We are more than happy to provide further clarifications.
> > >
> > > Thank you again for your careful review and valuable suggestions!
> > >
> > > Best,
> > >
> > > Authors

---

### Official Review · Reviewer_h9M6 · 2024-10-18

**Soundness:** 3
**Presentation:** 3
**Contribution:** 3
**Rating:** 6
**Confidence:** 5

**Summary:**

MMEvol addresses data quality and diversity challenges by proposing an iterative evolution of image-text instruction data. Starting with SEED-163K, it expands instruction types, enhances visual reasoning, and strengthens fine-grained perception and cognitive abilities.

**Strengths:**

This work proposes an image-text instruction evolution framework, Evol-Instruct, to enhance the quality and quantity of instruction data. Using three distinct evolution methods and instruction elimination to remove harmful data, the approach increases the complexity and diversity of a limited seed dataset. After three rounds of evolution, the resulting data trains a new model that achieves SOTA performance across various benchmarks.

**Weaknesses:**

- Evolving multimodal dataset makes sense and is so interesting but actual performnace improvements are too marginal in my perspective: 2~3% because evaluating language ability can improve large margin if it is really contrbutional approach.

- Experiments should be compared for fair comparison where same architecture, same dataset to provide the effectiveness of MMEvol.

- What kind of dataset samples are more effective to be applited by MMEvol like Math, code, or anyhting?

---

I will keep my score becuase it seems improvments are marginal than what I've expected (avg. 5%p~10%p).

**Questions:**

Refer to Weaknesses.

---

> ### Author Response · Authors · 2024-11-21
>
> Thanks for the valuable and encouraging comments! Our point-by-point responses to the reviewer's mentioned concerns are provided as follows.
>
> > **W 1**:  Evolving multimodal dataset makes sense and is so interesting but actual performnace improvements are too marginal in my perspective: 2~3% because evaluating language ability can improve large margin if it is really contrbutional approach.
>
> **Response:**
>
> We emphasize our 2-3% performance enhancement from the following points:
>
> 1. The data we evolved is relatively limited. The 163K dataset constitutes only 15% of the total dataset (1.1M).
>
> 2. The rewriting model we employed, GPT-4o-mini, is cost-effective but not optimal. If we utilize a more advanced open-source model such as Qwen2VL with 72 billion parameters, we could achieve further improvements.
>
> 3. In contrast to Cambrian's dataset of 7 million, which includes an additional  6 million high-quality instruction data, we have achieved better results using only 480K data points (8%) evolved from a limited seed dataset.
>
> Considering these three points, the fact that we achieved a 2-3% overall performance improvement with such a small amount of evolved data is quite acceptable. Should we scale up our efforts and use a more powerful  MLLM, the results would be very promising.
>
> | Data             | MMStar | MathVista$^M$ | MME$^C$ | AI2D | HallBench | MMMU$^V$ | RWQA | AVG. |
> | ---------------- | ------ | ------------- | ------- | ---- | --------- | -------- | ---- | ---- |
> | GPT4o-mini (3K)  | 37.9   | 26.1          | 31.3    | 55.1 | 43.8      | 35.8     | 53.2 | 40.5 |
> | Qwen2VL-72B (3K) | 39.1   | 27.9          | 33.1    | 57.8 | 46.4      | 36.9     | 46.9 | 41.2 |
>
> > **W 2**:  Experiments should be compared for fair comparison where same architecture, same dataset to provide the effectiveness of MMEvol.
>
> **Response:**
>
> To conduct a fair comparison, we randomly downsampled the same number of data points from MMEvol for a rigorous comparative experiment. The results of the experiment are presented in the table below. Under the same data volume and model architecture, MMEvol-8B achieved an average improvement of approximately 2.7 points, demonstrating the effectiveness of our approach.
>
> | Seed          | IT   | MMStar | MathVista$^M$ | MME$^C$ | AI2D | HallBench | MMMU$^V$ | RWQA | MIA  | BLINK | MMSInst | AVG. |
> | ------------- | ---- | ------ | ------------- | ------- | ---- | --------- | -------- | ---- | ---- | ----- | ------- | ---- |
> | LLaVA-Next-8B | 0.8M | 43.9   | 31.5          | 44.6    | 69.9 | 52.3      | 41.7     | 60.1 | 65.1 | 43.5  | 25.6    | 47.8 |
> | MMEvol-8B     | 0.8M | 48.4   | 41.3          | 45.9    | 71.3 | 47.6      | 40.2     | 61.2 | 72.6 | 45.2  | 30.9    | 50.5 |
>
> > **W 3**:  What kind of dataset samples are more effective to be applited by MMEvol like Math, code, or anyhting?
>
> **Response:**
>
> Thank you for your insightful concern. As we are solely focused on the textual portion of the instructional data during the evolutionary process, without supplementing new images, the text-centric multimodal instructional data type demonstrates greater efficiency and a higher success rate in terms of evolution. For instance, the success rate of evolutionary processes is significantly higher for types of data related to code and creative tasks. In contrast, for scientific diagrams (such as those in chemistry, physics, and mathematics), the success rate of evolution is relatively lower due to constraints imposed by the types and quantities of images available.
>
>
>
> Thank you again for your insightful comments.  If you have other comments, we are happy to address them to polish this work. We look forward to contributing to the development of both the Multi-Modal research and the open-source community.

---

> ### Author Response · Authors · 2024-11-22
>
> Dear Reviewer h9M6:
>
> We hope this message finds you well. We deeply appreciate your thoughtful feedback and the attention you’ve given to our manuscript. All concerns have been thoroughly addressed, and we wish to invite you to review the manuscript once more. With the deadline approaching, we would be grateful if you could confirm that all uncertainties have been resolved. We are ready to assist you with any further clarifications.
>
> Thanks for your cooperation.
>
> Kind regards,
>
> Authors

---

> ### Author Response · Authors · 2024-11-24
>
> Dear Reviewer h9M6,
>
> We want to express our gratitude for your thorough review and helpful comments on our manuscript. We've diligently worked on incorporating your suggestions and believe the revised version is much stronger. The deadline for discussions is approaching, and we would appreciate your feedback on our revisions at your earliest convenience. Your thoughtful evaluation is crucial to us.
>
> Thank you for your understanding and assistance.
>
> Kind regards,
>
> Authors

---

> ### Author Response · Authors · 2024-11-25
>
> Dear Reviewer h9M6,
>
> We are truly grateful for the thoughtful review you provided. We have taken all your feedback into consideration and revised the paper accordingly. Could you possibly re-evaluate our submission given the updates we have made? Your further feedback would be greatly appreciated, and we are prepared to clarify any remaining points of confusion.
>
> Many thanks,
>
> Authors

---

> ### Author Response · Authors · 2024-12-01
>
> Dear Reviewer h9M6,
>
> Your insights have been invaluable in refining our work, and we have diligently addressed each of your comments. As we approach the discussion deadline, we kindly ask if you could reassess our revised manuscript. We are more than willing to engage in further dialogue to ensure all your concerns are fully resolved.
>
> Thank you for your attention to this matter.
>
> Kind regards,
>
> Authors

---

### Official Review · Reviewer_pN7Z · 2024-10-29

**Soundness:** 3
**Presentation:** 3
**Contribution:** 3
**Rating:** 6
**Confidence:** 4

**Summary:**

In this paper, the authors propose MMEvol, a novel multimodal instruction data evolution framework that augments existing multi-modal training data with better diversity and complexity. The experiment results show that the proposed method works well and helps the MLLMs get better performance

**Strengths:**

Please refer to Questions

**Weaknesses:**

Please refer to Questions

**Questions:**

### Strength
1. The paper is well-written and easy to follow
2. The proposed idea is novel and seems to work well

### Weakness
1. My main concern is the comparison fairness in Table 2. The LLaVA-Next baseline uses the seed-set (163K data),  while the MMEvol uses an augmented set with more data(447K additional data). This leads to an unfair comparison, as in most cases, more data leads to better performance. This problem also exists in Table 1. The unfair comparison hinders the understanding of the actual effectiveness of the proposed method and I think it is easy to become fairer as the seed-set is sampled from existing open-source datasets and can easily be scaled up to a similar size with the augmented one.

2. Both the evolution and elimination are realized by the same model (GPT4o-mini). Is the model capable of finding out bad cases generated by itself? Further, I'm wondering about the fail rate and elimination rate of the proposed method.

3. Following 2, the proposed method evolute the instruction multiple times, will this lead to the error accumulation problem?

4. The paper focuses on improving the training data quality, while the provided example is quite limited. More data samples will help better evaluate the data quality.

I like the proposed idea and give it a 6: marginally above the acceptance threshold. But there are still some unclear problems I mentioned above. I will adjust the final score based on the rebuttal.

---

> ### Author Response · Authors · 2024-11-21
>
> Thanks for the valuable and encouraging comments! Our point-by-point responses to the reviewer's mentioned concerns are provided as follows.
>
> > **W 1**: This leads to an unfair comparison, as in most cases, more data leads to better performance.
>
> **Response:**
>
> We sincerely appreciate the detailed review comments. To ensure a fair comparison, we present the following experimental results. We downsampled MMevol to 0.8M and conducted a comparative experiment using LLaVA-Next. As shown in the table below, MMevol achieved an overall improvement of 2.7, demonstrating the effectiveness of the method.
>
> | Seed          | IT   | MMStar | MathVista$^M$ | MME$^C$ | AI2D | HallBench | MMMU$^V$ | RWQA | MIA  | BLINK | MMSInst | AVG. |
> | ------------- | ---- | ------ | ------------- | ------- | ---- | --------- | -------- | ---- | ---- | ----- | ------- | ---- |
> | LLaVA-Next-8B | 0.8M | 43.9   | 31.5          | 44.6    | 69.9 | 52.3      | 41.7     | 60.1 | 65.1 | 43.5  | 25.6    | 47.8 |
> | MMEvol-8B     | 0.8M | 48.4   | 41.3          | 45.9    | 71.3 | 47.6      | 40.2     | 61.2 | 72.6 | 45.2  | 30.9    | 50.5 |
>
> > **W 2**:  Is the model capable of finding out bad cases generated by itself? Further, I'm wondering about the fail rate and elimination rate of the proposed method.
>
> **Response:**
>
> To investigate the reliability of the rewrites produced by GPT-4-o-mini, we conducted a manual evaluation of the data before and after the evolution process. Specifically, we first extracted 30 images of various types from the seed data to ensure diversity, keeping 5 relevant question-answer pairs for each image. Subsequently, we carried out the corresponding evolution in three different directions, ultimately obtaining 450 evolved question-answer pairs, which were then subject to scoring and filtering. The results were distributed among five experts for manual evaluation of the accuracy of the model evolution and the scoring filter. The data is summarized in the table below. From the table, it is evident that the average success rate of evolution using MLLM can reach 90%, while the accuracy of the scoring filter can achieve 94%, indicating the reliability of MMEovel. Additionally, we provide detailed scoring cases in Figure 15, highlighted in red.
>
> | data id           | expert | image categories                                           | FP-Evol (0-5) | I-Evol (0-5) | CR-Evol (0-5) | I-Elim (0-15)(450) |
> | ----------------- | ------ | ---------------------------------------------------------- | ------------- | ------------ | ------------- | ------------------ |
> | 0,1,3,4,5,6       | 0      | LandMark,OCR,Human&Clothes,Traffic,Living room,Sport       | 5,4,4,5,5,4   | 5,4,3,4,5,4  | 5,3,4,5,4,4   | 15,13,13,14, 13,14 |
> | 7,8,9,10,11,12    | 1      | Kitchen,Office supplies&Tools,Plants,Animal,Sport,LandMark | 5,5,4,5,4,4   | 5,4,5,5,4,4  | 5,5,4,4,5,4   | 14,15,13,15,14,13  |
> | 13,14,15,16,17,18 | 2      | Foods,LandMark,OCR,Human&Clothes,Traffic,Sport             | 4,4,3,5,4,5   | 5,4,4,4,4,5  | 4,5,5,4,5,5   | 14,14,15,13,14,15  |
> | 19,20,21,22,23,24 | 3      | Foods,Sport,LandMark,Office supplies&Tools,Plants,Traffic  | 3,4,5,5,5,4   | 3,4,5,5,5,5  | 5,5,5,5,5,5   | 13,15,14, 15,15,15 |
> | 25,26,27,28,29,30 | 4      | Animal,Sport,Traffic,Landmark,Sport,Office supplies&Tools  | 4,5,5,5,5,5   | 4,5,5,5,4,5  | 5,5,3,5,5,5   | 14,15,14,15,14,15  |
> |                   |        |                                                            | 89.3%         | 88.7%        | 92%           | 94.5%              |

---

> > ### Author Response · Authors · 2024-11-21
> >
> > > **W 3**: Following 2, the proposed method evolute the instruction multiple times, will this lead to the error accumulation problem?
> >
> > **Response:**
> >
> > Cumulative error is inevitable.However, it is possible to estimate the number of samples contributing to the cumulative error after three iterations as approximately $0.055^3$ × 160K, which amounts to around 30. This is a relatively small quantity (30 vs. 160K) and exerts a marginal impact on the quality of the evolved data.
> >
> > > **W 4**: The paper focuses on improving the training data quality, while the provided example is quite limited. More data samples will help better evaluate the data quality.
> >
> > **Response:**
> >
> > Thank you for your valuable suggestions. In the revised version, we have incorporated additional visual cases in Figures 21-23 (highlighted in red) to effectively illustrate the validity of our evolution.
> >
> > Thank you again for your insightful comments.  If you have other comments, we are happy to address them to polish this work. We look forward to contributing to the development of both the Multi-Modal research and the open-source community.

---

> ### Author Response · Authors · 2024-11-22
>
> Dear Reviewer pN7Z:
>
> We are truly grateful for your insightful comments and the guidance you provided during your review of our paper. We are pleased to inform you that we have addressed all points raised and have made significant improvements. As the discussion phase draws near, we kindly request your reevaluation at your earliest convenience. Should any questions remain, we are at your disposal to clarify them promptly.
>
> Thank you for your time and understanding.
>
> Sincerely,
>
> Authors

---

> ### Author Response · Authors · 2024-11-24
>
> Dear Reviewer pN7Z,
>
> Firstly, thank you for your meticulous review and constructive feedback. We have addressed all the points you raised and have clarified any ambiguities in the revised manuscript. As the deadline for discussions nears, we kindly ask if you could review our changes and provide any further comments. Your inputs are invaluable, and we're committed to ensuring the manuscript meets your standards.
>
> Thank you for your continued support.
>
> Best wishes,
>
> Authors

---

> ### Author Response · Authors · 2024-11-25
>
> Dear Reviewer pN7Z,
>
> Your insights have been invaluable in refining our work, and we have diligently addressed each of your comments. As we approach the discussion deadline, we kindly ask if you could reassess our revised manuscript. We are more than willing to engage in further dialogue to ensure all your concerns are fully resolved.
>
> Thank you for your attention to this matter.
>
> Kind regards,
>
> Authors

---

> > ### Comment · Reviewer_pN7Z · 2024-11-29
> >
> > Thanks for the detailed rebuttal, my main concern is solved and I keep my score as 6.

---

### Official Review · Reviewer_8Cfc · 2024-11-04

**Soundness:** 2
**Presentation:** 3
**Contribution:** 2
**Rating:** 5
**Confidence:** 4

**Summary:**

This paper addresses the lack of high-quality data in building stronger multimodal large language models and proposes a solution through the development of a multimodal instruction data evolution framework, MMEval. The framework iteratively improves data quality across three evolution stages: fine-grained perception, cognitive reasoning, and interactive evolution. The authors demonstrate the effectiveness of the framework by refining a new dataset called SEED-163K and training a large multimodal model on this refined dataset. Evaluation results across 13 benchmarks further validate the effectiveness of the proposed framework.

**Strengths:**

- The paper addresses the issue of data scarcity in instruction datasets for training multimodal large language models, presenting a well-motivated problem.
- The authors designed a sophisticated pipeline to enhance data quality, featuring three evolution stages and an instruction elimination stage.
- The authors evaluate the effectiveness of the three proposed evolution stages, as shown in Table 1.
- The performance of **MMEval** MLLMs in Table 2 appears promising.

**Weaknesses:**

- The entire data improvement framework relies on closed-source frontier models like GPT-4, which suggests a form of knowledge distillation from these models but may limit the ability to scale to larger datasets. Additionally, the strong dependence on models like GPT-4 reduces the framework's interpretability.
- The authors do not provide an analysis of the reliability of using GPT-4 as a data rewriter.
- The paper lacks a comparison of different prompts used in each evolution stage, leaving the impact of prompt templates on data refinement unclear.
- When comparing the results of **MMEval-8B** with **Cambrian-1 8B** in Table 2, although **MMEval-8B** shows overall improvements, it exhibits significant performance declines on key benchmarks like **MMMU**, **AI2D**, and **MMStar**.

**Questions:**

- How reliable is GPT-4 as a data rewriter? How can the quality of the rewritten data be evaluated?
- Is the rewritten output sensitive to changes in the prompt?
- What is the rationale behind the current prompt design? Have other prompt variations been compared?
- Why does **MMEval-8B** perform poorly on **MMMU**?

---

> ### Author Response · Authors · 2024-11-21
>
> Thanks for the valuable and encouraging comments! Our point-by-point responses to the reviewer's mentioned concerns are provided as follows.
>
> > **W 1**The entire data improvement framework relies on closed-source frontier models like GPT-4, which suggests a form of knowledge distillation from these models but may limit the ability to scale to larger datasets. Additionally, the strong dependence on models like GPT-4 reduces the framework's interpretability.
>
> **Response:**
>
> In this study, we employ the GPT-4o mini model as a cost-effective alternative to the GPT-4 for data evolution. Our results demonstrate that its performance is comparable to that of the latter one, while offering lower costs (**15K$ vs. 600$**). Additionally, the GPT-4o mini's performance aligns closely with established high-performance open-source alternatives, making it a favorable choice for our evolution. We present the results of 3K  data evolved using the open-source model Qwen2VL 72B, as illustrated in the table below.
>
> | Data           | MMStar | MathVista$^M$ | MME$^C$ | AI2D | HallBench | MMMU$^V$ | RWQA | AVG. |
> | -------------- | ------ | ------------- | ------- | ---- | --------- | -------- | ---- | ---- |
> | GPT4o-mini-3K  | 37.9   | 26.1          | 31.3    | 55.1 | 43.8      | 35.8     | 53.2 | 40.5 |
> | Qwen2VL-72B-3K | 39.1   | 27.9          | 33.1    | 57.8 | 46.4      | 36.9     | 46.9 | 41.2 |
>
> Compared to GPT-4o-mini, utilizing the more powerful open-source Qwen2VL 72B yields superior results, demonstrating the scalability and practicality of our approach.
>
> > **Q1&W 2**: The authors do not provide an analysis of the reliability of using GPT-4 as a data rewriter. How reliable is GPT-4 as a data rewriter? How can the quality of the rewritten data be evaluated?
>
> **Response:**
>
> To investigate the reliability of the rewrites produced by GPT-4-o-mini, we conducted a manual evaluation of the data before and after the evolution process. Specifically, we first extracted 30 images of various types from the seed data to ensure diversity, keeping 5 relevant question-answer pairs for each image. Subsequently, we carried out the corresponding evolution in three different directions, ultimately obtaining 450 evolved question-answer pairs, which were then subject to scoring and filtering. The results were distributed among five experts for manual evaluation of the accuracy of the model evolution and the scoring filter. The data is summarized in the table below. From the table, it is evident that the average success rate of evolution using MLLM can reach 90%, while the accuracy of the scoring filter can achieve 94%, indicating the reliability of MMEovel. Additionally, we provide detailed scoring cases in Figure 15, highlighted in red.
>
> |           data id | expert | image categories                                           | FP-Evol (0-5) | I-Evol (0-5) | CR-Evol (0-5) | I-Elim (0-15)(450) |
> | ----------------: | ------ | ---------------------------------------------------------- | ------------- | ------------ | ------------- | ------------------ |
> |       0,1,3,4,5,6 | 0      | LandMark,OCR,Human&Clothes,Traffic,Living room,Sport       | 5,4,4,5,5,4   | 5,4,3,4,5,4  | 5,3,4,5,4,4   | 15,13,13,14, 13,14 |
> |    7,8,9,10,11,12 | 1      | Kitchen,Office supplies&Tools,Plants,Animal,Sport,LandMark | 5,5,4,5,4,4   | 5,4,5,5,4,4  | 5,5,4,4,5,4   | 14,15,13,15,14,13  |
> | 13,14,15,16,17,18 | 2      | Foods,LandMark,OCR,Human&Clothes,Traffic,Sport             | 4,4,3,5,4,5   | 5,4,4,4,4,5  | 4,5,5,4,5,5   | 14,14,15,13,14,15  |
> | 19,20,21,22,23,24 | 3      | Foods,Sport,LandMark,Office supplies&Tools,Plants,Traffic  | 3,4,5,5,5,4   | 3,4,5,5,5,5  | 5,5,5,5,5,5   | 13,15,14, 15,15,15 |
> | 25,26,27,28,29,30 | 4      | Animal,Sport,Traffic,Landmark,Sport,Office supplies&Tools  | 4,5,5,5,5,5   | 4,5,5,5,4,5  | 5,5,3,5,5,5   | 14,15,14,15,14,15  |
> |                   |        |                                                            | 89.3%         | 88.7%        | 92%           | 94.5%              |

---

> > ### Author Response · Authors · 2024-11-21
> >
> > > **Q2&W 3&Q3**: The paper lacks a comparison of different prompts used in each evolution stage, leaving the impact of prompt templates on data refinement unclear. Is the rewritten output sensitive to changes in the prompt? What is the rationale behind the current prompt design? Have other prompt variations been compared?
> >
> > **Response:**
> >
> > Base Version of Prompt as below:
> >
> > FP-Evol: I want you act as a Q&A Creator. Your objective is to draw inspiration from the given Q&A to create a brand new created Q&A.
> >
> > I-Evol: I want you act as a Q&A Rewriter. Your objective is to rewrite a given Q&A into a more complex form to meet real word interactive demand.
> >
> > CR-Evol: I want you act as a Q&A Rewriter. Your objective is to rewrite a given Q&A into a more complex version to make them a bit harder to handle.
> >
> > To verify the effectiveness of our prompt design, we conducted an ablation study using the base prompt on 1K seed data, while maintaining equal evolutionary probabilities across three fixed directions. Additionally, we provided supplementary visual results using the base prompt in Figure 11 of the paper, highlighted in red.
> >
> > | FP-Evol      | I-Evol       | CR-Evol      | MMStar | MathVista$^M$ | MME$^C$ | AI2D | HallBench | MMMU$^V$ | RWQA | AVG. |
> > | ------------ | ------------ | ------------ | ------ | ------------- | ------- | ---- | --------- | -------- | ---- | ---- |
> > | $\checkmark$ | $\checkmark$ | $\checkmark$ | 34.7   | 25.7          | 29.9    | 54.1 | 42.1      | 35.5     | 49.8 | 38.8 |
> > | $\checkmark$ | $\checkmark$ | $\times$     | 35.7   | 25.9          | 30.3    | 54.8 | 42.9      | 35.2     | 51.2 | 39.4 |
> > | $\checkmark$ | $\times$     | $\times$     | 36.5   | 25.4          | 30.8    | 55.0 | 43.6      | 35.4     | 52.4 | 39.9 |
> > | $\times$     | $\times$     | $\times$     | 37.9   | 26.1          | 31.3    | 55.1 | 43.8      | 35.8     | 53.2 | 40.5 |
> >
> > The symbol $\checkmark$ indicates that during the evolutionary process, the prompt has been replaced with the base version, as demonstrated in the table. Utilizing our meticulously designed prompts significantly enhances the diversity and complexity of the data, thereby making the evolutionary process more efficient.
> >
> > > **W 4&Q4**: When comparing the results of **MMEval-8B** with **Cambrian-1 8B** in Table 2, although **MMEval-8B** shows overall improvements, it exhibits significant performance declines on key benchmarks like **MMMU**, **AI2D**, and **MMStar**. Why does **MMEval-8B** perform poorly on **MMMU**?
> >
> > **Response:**
> >
> > Compared to Cambrian-1 8B, even with the utilization of only 8% additional data (480K vs. 6M), our model demonstrates comparable performance on key benchmarks such as **MMMU**, **AI2D**, and **MMStar**. Furthermore, continual enhancement of image quantity during the evolutionary process (for instance, through a simple rewriting of $VILA^2$ [1] rather than multiple iterations) would yield significant improvements on the MMMU benchmark. This suggests that for evaluative datasets like MMMU, which consists of college-level textbook questions, relying solely on limited image datasets for instruction evolution may yield relatively modest enhancements compared to text-centered capabilities if there is insufficient new image data available. However, it is important to note that the addition of image data and instruction evolution can be synergistically combined, resulting in more substantial improvements overall.
> >
> > [1] Yunhao Fang, et al."$VILA^2$: VILA Augmented VILA"  *arXiv preprint arXiv:2407.17453* (2024).
> >
> >
> >
> >
> >
> > Thank you again for your insightful comments.  If you have other comments, we are happy to address them to polish this work. We look forward to contributing to the development of both the Multi-Modal research and the open-source community.

---

> > > ### Author Response · Authors · 2024-11-22
> > >
> > > Dear Reviewer 8Cfc:
> > >
> > > Thank you immensely for your valuable feedback on our manuscript. We've worked diligently to incorporate your suggestions and make necessary revisions. With the review timeline approaching, we kindly ask if you could spare a moment to re-evaluate the updated document. Please let us know if there is anything else you need from our end for clarification.
> > >
> > > We truly appreciate your cooperation and continued support.
> > >
> > > Warm regards,
> > >
> > > Authors

---

> ### Author Response · Authors · 2024-11-24
>
> Dear Reviewer 8Cfc,
>
> I hope this email finds you in good spirits. We are grateful for your detailed evaluation and have worked hard to address your concerns. With the discussion deadline fast approaching, we would appreciate it if you could revisit the manuscript to see if the adjustments align with your expectations. Your expertise continues to play a pivotal role in refining our work.
>
> Thank you once again for your dedication.
>
> Sincerely,
>
> Authors

---

> ### Author Response · Authors · 2024-11-25
>
> Dear Reviewer 8Cfc,
>
> Thank you very much for your detailed review and constructive feedback. We have carefully revised the manuscript to resolve the issues you've pointed out. With the discussion deadline approaching, we would be grateful if you could review our changes. Should you require any further clarifications, please let us know, and we will gladly provide them promptly.
>
> Best wishes,
>
> Authors

---

> > ### Comment · Reviewer_8Cfc · 2024-11-27
> > **Reponses to Authors' Comments**
> >
> > I sincerely thank the authors for their comprehensive responses to my concerns. However, I still tend to maintain my initial score for the following reasons:
> > 1. Scalability Concerns: The authors attempt to demonstrate the scalability of the proposed method by conducting an additional experiment using an open-source state-of-the-art (SoTA) method as a data rewriter. However, my concern remains that regardless of whether the rewriter is open-source or closed-source, it inevitably serves as a performance upper bound for the proposed method. This implies that surpassing the rewriter's performance is unattainable if the method merely distills knowledge from it as a teacher. I consider this a significant limitation of the paper, yet the authors neither address it adequately nor mention it explicitly in the manuscript.
> > 2. Scalability and Practicality: The authors claim that their method is both scalable and practical. However, I find no concrete evidence supporting this assertion, such as experiments involving larger datasets or models. This raises the question: what exactly is meant by scalability in this context?
> > 3. Prompt Sensitivity: Regarding my concern about prompt sensitivity, the authors provide a comparison between their prompt and a few naive alternatives. However, they fail to address critical aspects, such as the rationale behind their specific prompt design or whether the rewritten outputs are sensitive to variations in the prompt. Considering that the paper focuses on designing a data refinement pipeline where the prompt plays a pivotal role, the lack of insights in this area significantly weakens its contributions.

---

> > > ### Author Response · Authors · 2024-11-28
> > >
> > > Thanks for the valuable and insightful comments! Our point-by-point responses to the reviewer's mentioned concerns are provided as follows.
> > >
> > > > **Q 1**: Scalability Concerns:
> > >
> > > **Response:**
> > >
> > > Thank you for your valuable suggestion. Your main concern pertains to whether the data evolution in MMEvol can surpass the upper bound. Here, we provide further clarification. First, in the open-source multimodal community, there are numerous fully open-source works, such as LLaVA-OneVision [1] and Molmo [2]. These works have utilized a large amount of multimodal data constructed using closed-source multimodal teacher models, ultimately achieving results superior to those of the teacher models. Similarly, self-evolution work like VILA2 [3] have successfully surpassed the upper bound on MLLM performance through multiple cycles of simple rewriting on extensive datasets. These works collectively validate the efficacy of using multimodal teacher models to construct data that eventually trains stronger multimodal models, outperforming the original teacher models. Consequently, MMEvol emerges as a promising approach, offering a more efficient method for multimodal data construction and achieving this with lesser data than prior efforts.
> > >
> > > Secondly, the rapid progress in instruction evolution is grounded in iterative comparison during synthetic data generation, guiding data synthesis through clear directions in complexity and diversity. This approach generates more complex and diverse data, eventually outperforming the teacher models. Such methodologies have already been validated in domains like code [4], mathematics [5], and text [6]. Our approach successfully applies the principles of instruction evolution to the multimodal domain.
> > >
> > > Lastly, we further elucidate that the breakthrough of the upper bound is a conclusion drawn from previous works [1,2,3]. The core contribution of MMEvol lies in providing a more efficient method for generating multimodal data. It can quickly construct a substantial amount of high-quality data from a limited base, intentionally enhancing diversity and complexity. Compared to simple rewriting methods like MIMIC-IT [7], ALLaVA [8], and MM-Instruct [9], which lack clear objectives, our method demonstrates higher efficiency and better outcomes. As shown in the table below, MMEvol surpassed the teacher model on both HallBench and POPE using only 15% of full data (1.1M) for evolution.
> > >
> > > | Model            | HallBench | POPE     |
> > > | ---------------- | --------- | -------- |
> > > | GPT4o-mini (API) | 61.9      | 86.1     |
> > > | MMEvol           | **64.1**  | **87.8** |
> > >
> > > > **Q 2**: Scalability and Practicality
> > >
> > > **Response:**
> > >
> > > Thank you very much for your detailed review. We have included the missing scalability study below. Due to the absence of models at the scale of Qwen2 and LLaMA3 13B, we have chosen Vicuna 1.5 to conduct the ablation experiments.
> > >
> > > | Model                   | MMStar   | MathVista$^M$ | MME$^C$  | AI2D     | HallBench | MMMU$^V$ | RWQA     | AVG.     |
> > > | ----------------------- | -------- | ------------- | -------- | -------- | --------- | -------- | -------- | -------- |
> > > | Vicuna-7B (seed 3K)     | 28.7     | 20.7          | 22.9     | 38.9     | 39.6      | 29.3     | 43.2     | 31.9     |
> > > | Vicuna-7B (evolved 3K)  | 30.9     | 22.0          | 25.6     | 41.2     | 42.3      | 31.6     | 46.3     | 34.3     |
> > > | Vicuna-7B (evolved 6K)  | 31.4     | 23.2          | 28.6     | 43.5     | 44.6      | 32.3     | 48.1     | 36.0     |
> > > | Vicuna-7B (evolved 9K)  | 31.9     | 24.0          | 31.1     | 44.7     | 47.4      | 33.8     | 50.3     | 37.6     |
> > > | Vicuna-13B (evolved 9K) | **34.6** | **26.1**      | **34.5** | **50.6** | **52.3**  | **36.1** | **54.5** | **41.3** |
> > >
> > > As shown in the table, the model's multimodal capabilities can be further enhanced with increasing data volume and model scale. Larger-scale models exhibit greater performance improvements when trained on our high-quality data.
> > >
> > > [1] Bo Li, et al. LLaVA-OneVision: Easy Visual Task Transfer  (2024).
> > >
> > > [2] Matt Deitke, et al. Molmo and PixMo: Open Weights and Open Data for State-of-the-Art Multimodal Models   (2024).
> > >
> > > [3] Yunhao Fang, et al. $VILA^2$: VILA Augmented VILA   (2024).
> > >
> > > [4] Ziyang Luo, et al. WizardCoder: Empowering Code Large Language Models with Evol-Instruct   (2023).
> > >
> > > [5] Haipeng Luo, et al. WizardMath: Empowering Mathematical Reasoning for Large Language Models via Reinforced Evol-Instruct   (2023).
> > >
> > > [6] Can Xu, et al. WizardLM: Empowering Large Language Models to Follow Complex Instructions   (2023).
> > >
> > > [7] Li et al., MIMIC-IT: Multi-Modal In-Context Instruction Tuning, 2023.
> > >
> > > [8] Guiming et al., ALLaVA: Harnessing GPT4V-Synthesized Data for Lite Vision-Language Models, 2024.
> > >
> > > [9] Liu et al., MM-Instruct: Generated Visual Instructions for Large Multimodal Model Alignment, 2024.

---

> > > > ### Author Response · Authors · 2024-11-28
> > > >
> > > > > **Q 3**: Prompt Sensitivity
> > > >
> > > > **Response:**
> > > >
> > > > Thank you for your valuable suggestion. We are pleased to further elucidate the design rationale of our prompt, as elaborated in our paper. Initially, we identified three common deficiencies in existing multimodal data. To address these issues, we devised three evolutionary strategies tailored to each deficiency, forming a meaningful motivation for our approach. However, the core challenge in evolving multimodal instructions lies in assessing the quality of multimodal data, including its degree of complexity and diversity. It is only by quantifying these attributes that we can effectively enhance the data's complexity and diversity.
> > > >
> > > > To tackle this challenge, we adopted the classification scheme outlined in Cambrain-I [10], which categorizes multimodal capabilities into visually-centric and language-centric atomic capabilities. Each multimodal problem is decomposed into a combination of atomic capabilities and atomic goals, with the introduction of a visually-centric visual operation chain to measure the level of reasoning complexity. Through these designs, we can assess and specifically direct the evolution of multimodal data's complexity and diversity.
> > > >
> > > > We adhere to three principles to maximize the length of the visual operation chain, increase data on atomic capabilities, and diversify atomic goal types. Ultimately, this approach yielded the most concise prompts across the three respective directions and successfully drove the evolution of multimodal instructions. Our prompt design rationale strictly adheres to the principles of making instructional evolution feasible and efficient through minimalistic design.
> > > >
> > > > [10] Shengbang , et al. Cambrian-1: A Fully Open, Vision-Centric Exploration of Multimodal LLMs (2024).
> > > >
> > > >
> > > >
> > > > We appreciate your response and will actively participate in the discussion. If our feedback solves your concerns or you have other concerns, kindly let us know. We will do our best to address them for you and enhance this work.

---

### Official Review · Reviewer_1KdT · 2024-11-11

**Soundness:** 3
**Presentation:** 3
**Contribution:** 3
**Rating:** 6
**Confidence:** 4

**Summary:**

This paper introduce a method to systematically evolve seed instruction fine-tuning data for VLM to larger scale and enhance vision centric capabilities for fine-tuned visual understanding, visual reasoning and human interactive capabilities. The enhanced evolution dataset show signiificant improvement on a wide range of perception and reasoning dataset.

**Strengths:**

The problems this paper tries to address are spot on, current VLMs is limited in terms of complex visual reasoning, natural interaction with human, etc. The method to evolve current instruction fine-tune data to enhance these capabilities is one of interesting direction to improve the model’s capabilities, as shown in the benchmark results.

The authors mention data and code will be released, and I think it’ll be a good contribution to the community to set a higher baseline for LLAVA style model and maybe even beyond.

**Weaknesses:**

My major concern is lack of technical clarity and minor concern is insufficient evaluation.

First, the prompt is clear in Fig 4 - Fig 7, can you explictly describe what model, commercial API or other method is used to generate the evolutoin instruction data from the prompt templates?

Second, it is mentioned pseudo function calling is used for visual reasoning evolution, can you describe the setup of function call, the model or template used to generate the function call, and how it is integrataed to the visual reason evolution process?

Third, it is mentioned MLLM is used for rewriting, can you describe details on what MLLM is used, how it is used (prompt used), evalution or abalation of the significance of this rewriting step?

On the evaluation side, I think a more comprehensive evaluation on different capabilities of the model and some more ablation would provide more insights to the method and data.

Firrst, the seed dataset and one of the vision-centric capabilities is OCR, while there is few OCR related benchmark results, more results on OCRBench, ChratQA, DocVQA, TextVQA would be very insightful.

Second, more ablation on ratio of the three evolution methods in each round, how to decide and eliminate failed evolution, what’s the success/fail ratio for each round, what is the model quality gain for reach round, etc. would be informative.

Third, in comparison with other methods, InternVL2-8b (releaesd 2024/07) and Qwen2-VL-7b (released 2024/08) should be included in Table2 under weight open source section.

**Questions:**

Addressing questions w.r.t. Technical clarify and evaluation in weakness session would impact my final judgement of the paper. The following questions are nice-to-have discussion which might not impact final score.

Enhancing vision centric capabilities in fine-grained object, CoT, interaction seems to be effective in LLAVA style VLM, do you think it is because the pretrained ViT / LLM lacks these capabilities in the first place, or pretrained models already have learned enough knowledge but somehow forget it with poor instruction data during fine-tuning? Do you think this data will help other pretrained VLMs trained with tens of billions of image/text tokens?

---

> ### Author Response · Authors · 2024-11-21
>
> Thanks for your professional and careful review. We respond to your concerns or questions as follows.
>
> > **W 1**: First, the prompt is clear in Fig 4 - Fig 7, can you explictly describe what model, commercial API or other method is used to generate the evolutoin instruction data from the prompt templates?
>
> **Response:**
>
> We utilized GPT-4o-mini, and as mentioned in Appendix C and the Limitation Section, we will include a clearer description in the revised version of the main paper.
>
> > **W 2**: It is mentioned pseudo function calling is used for visual reasoning evolution, can you describe the setup of function call, the model or template used to generate the function call, and how it is integrataed to the visual reason evolution process?
>
> **Response:**
>
> We have provided the following example regarding Figure 3 of the paper and offered a further explanation of the evolution of cognitive reasoning.
>
> **Seed Sample:**
>
> ```json
> {
>   "objects": ["window", "couch", "vase", ...],
>   "skills": ["Grounding Ability", "Context Understanding Ability", ...],
>   "format": "Normal",
>   "question": "Which is closer to the window, the couch or the vase?",
>   "steps": [],
>   "answer": "The couch."
> }
> ```
>
> **Evolved Sample:**
>
> ```json
> {
>   "objects": ["window", "couch", "vase", ...],
>   "skills": ["Grounding Ability", "Context Understanding Ability", ...],
>   "format": "Complex reasoning",
>   "question": "Which is closer to the window, the couch or the vase?",
>   "steps": [
>     {
>       "manipulation": "grounding_1(window) -> bbx_1",
>       "description": "Locate the window in the room and return its bounding box as bbx_1."
>     },
>     {
>       "manipulation": "grounding_2(couch) -> bbx_2",
>       "description": "Locate the couch in the room and return its bounding box as bbx_2."
>     },
>     {
>       "manipulation": "grounding_3(vase) -> bbx_3",
>       "description": "Locate the vase in the room and return its bounding box as bbx_3."
>     },
>     {
>       "manipulation": "Calculate([window, couch/vase]) -> res_i",
>       "description": "Calculate the spatial distance between the window and the couch/vase."
>     }
>   ],
>   "answer": "First, based on the detected bounding boxes of the window [0.2, 0.23, 0.57, 0.4] and the couch [0.17, 0.43, 0.83, 0.79], it can be inferred that the couch is positioned below the window. Further, according to the location of the vase [0.5, 0.51, 0.58, 0.72], it is located near the center of the room. Therefore, the couch is closer to the window."
> }
> ```
>
> In this example, we instruct the model to generate a series of reasoning steps (the "steps" field) using pseudo function calls. These steps simulate operations such as grounding objects and calculating spatial relationships. By incorporating these steps into the evolution process, we enhance the model's ability to perform complex visual reasoning.  It is important to note that we achieve both processes simultaneously by incorporating requirements in the prompt, which ultimately yields more complex instructions and enhances the model's reasoning capabilities.
>
> > **W 3**: Third, it is mentioned MLLM is used for rewriting, can you describe details on what MLLM is used, how it is used (prompt used), evalution or abalation of the significance of this rewriting step?
>
> **Response:**
>
> Kindly refer to W1&W2 and the corresponding replies. We utilized the GPT-4o-mini model and formatted the instruction data based directly on the prompts, which were then fed into the model to generate and parse the evolved data. To investigate the impact of removing the chain of reasoning on inductive rewriting, we also conducted the following ablation experiments on 3K evolved data.
>
> | CR-Evol        | MMStar | MathVista$^M$ | MME$^C$ | AI2D | HallBench | MMMU$^V$ | RWQA | AVG. |
> | -------------- | ------ | ------------- | ------- | ---- | --------- | -------- | ---- | ---- |
> | w/o rewritting | 37.2   | 25.4          | 30.1    | 54.7 | 43.6      | 34.2     | 51.3 | 39.5 |
> | w /rewritting  | 37.9   | 26.1          | 31.3    | 55.1 | 43.8      | 35.8     | 53.2 | 40.5 |
>
> As we can see from the table, the induction of reasoning chains and rewriting requirements into the answer are crucial.

---

> > ### Author Response · Authors · 2024-11-21
> >
> > > **W 4**: Firrst, the seed dataset and one of the vision-centric capabilities is OCR, while there is few OCR related benchmark results, more results on OCRBench, ChratQA, DocVQA, TextVQA would be very insightful.
> >
> > **Response:**
> >
> > We sincerely appreciate the your valuable feedback. We have included additional evaluations related to OCR, as shown in the table below. The results demonstrate that MMEvol significantly enhances the OCR capabilities of MLLM compared with the Baseline.
> >
> > | Model     | OCRBench | ChartQA | DocVQA | TextVQA |
> > | --------- | -------- | ------- | ------ | ------- |
> > | SEED-8B   | 57.3     | 70.4    | 79.2   | 69.8    |
> > | MMEvol-8B | 61.2     | 74.6    | 84.6   | 74.6    |
> >
> > > **W 5**: Second, more ablation on ratio of the three evolution methods in each round, how to decide and eliminate failed evolution, what’s the success/fail ratio for each round, what is the model quality gain for reach round, etc. would be informative.
> >
> > **Response:**
> >
> > Thank you to your detail review. As shown in Fig.7 in the main paper, we prompt the GPT-4o mini to analyze the evolutionary gain and complexity levels of generated instruction data. As for data elimination, each generated sample will be rated on a difficulty scale of 1 to 10 according to the evaluation criteria in Fig. 7, samples that do not demonstrate significant evolutionary advancement, *i.e.*, “improved=False” or “improved=True” while receiving a score below 6, will be eliminated from further consideration. As suggested, we have tallied the failure rates for each round as shown in the table below.
> >
> > | Round-1 | Round-2 | Round-3 |
> > | ------- | ------- | ------- |
> > | 26%     | 24%     | 20%     |
> >
> > Moreover, we conducted ablation experiments on the ratios of the three evolutionary directions during the evolution process of the 1K data, as shown in the table below. The results indicate that when the ratios of the three evolutionary directions are equal, the highest average performance can be achieved, thereby demonstrating that all three directions are equally important for the diversity and complexity of the evolutionary instruction data.
> >
> > | FP-Evol | I-Evol | CR-Evol | MMStar | MathVista$^M$ | MME$^C$ | AI2D | HallBench | MMMU$^V$ | RWQA | AVG. |
> > | ------- | ------ | ------- | ------ | ------------- | ------- | ---- | --------- | -------- | ---- | ---- |
> > | 2/3     | 1/6    | 1/6     | 36.9   | 26.3          | 31.0    | 54.0 | 44.8      | 34.4     | 51.4 | 39.8 |
> > | 1/6     | 2/3    | 1/6     | 34.3   | 25.4          | 29.2    | 53.2 | 43.5      | 35.8     | 52.6 | 39.2 |
> > | 1/6     | 1/6    | 2/3     | 36.3   | 26.7          | 32.5    | 54.3 | 44.0      | 35.2     | 51.1 | 40.0 |
> > | 1/3     | 1/3    | 1/3     | 37.9   | 26.1          | 31.3    | 55.1 | 43.8      | 35.8     | 53.2 | 40.5 |
> >
> > > **W 6**: Third, in comparison with other methods, InternVL2-8b (releaesd 2024/07) and Qwen2-VL-7b (released 2024/08) should be included in Table2 under weight open source section.
> >
> > **Response:**
> >
> > We have revised the missing citation and made changes in Table 2, highlighting them in red. We appreciate your valuable feedback, which has contributed to the improvement of our work.
> >
> > > **Q 1**: it is because the pretrained ViT / LLM lacks these capabilities in the first place, or pretrained models already have learned enough knowledge but somehow forget it with poor instruction data during fine-tuning? Do you think this data will help other pretrained VLMs trained with tens of billions of image/text tokens?
> >
> > **Response:**
> >
> > This is a very interesting question. We believe the latter is correct. The pre-trained model has already acquired sufficient general knowledge and coarse-grained alignment. Therefore, during the supervised fine-tuning phase, it is only necessary to utilize high-quality, fine-grained instructional data to effectively activate this knowledge, which will lead to improved performance. We think our data can assist other pre-trained Vision-Language Models (VLMs) in achieving better training performance. Subsequently, we will release approximately 10 million high-quality evolutionary data points generated using MMEvol to support the open-source community in building more robust fully open-source Multimodal Large Language Models (MLLMs).
> >
> >
> >
> > Thank you again for your insightful comments.  If you have other comments, we are happy to address them to polish this work. We look forward to contributing to the development of both the Multi-Modal research and the open-source community.

---

> > > ### Author Response · Authors · 2024-11-22
> > >
> > > Dear Reviewer 1KdT:
> > >
> > > We greatly appreciate the time and effort you dedicated to reviewing our paper. We have carefully addressed all your insightful suggestions and clarified any ambiguous points to improve our work. As the deadline for the discussion is nearing, could you kindly reconsider your evaluation based on the revised version? We are open to any further queries you might have and are eager to provide any additional information needed.
> > >
> > > Thank you for your understanding and support.
> > >
> > > Best regards,
> > >
> > > Authors

---

> ### Author Response · Authors · 2024-11-24
>
> Dear Reviewer 1KdT,
>
> Thank you for contributing your time and expertise to review our manuscript. We've taken your insightful comments seriously and amended the paper accordingly. As we approach the discussion deadline, we are eager to hear your thoughts on the revised version. Should any points still need clarification, we're ready to assist promptly.
>
> Thank you for your indispensable guidance.
>
> With gratitude,
>
> Authors

---

> ### Author Response · Authors · 2024-11-25
>
> Dear Reviewer 1KdT,
>
> We deeply appreciate the time and effort you have invested in reviewing our paper. We have thoroughly addressed your valuable comments and made the necessary revisions. Could you kindly re-evaluate our manuscript at your earliest convenience? We are more than willing to discuss any remaining concerns you might have.
>
> Thank you for your understanding and cooperation.
>
> Best regards,
>
> Authors

---

> ### Author Response · Authors · 2024-12-01
>
> Dear Reviewer 1KdT,
>
> We hope this message finds you well.
>
> We have carefully addressed your questions and concerns in the rebuttal, including conducting additional experiments and providing detailed clarifications.
>
> As the rebuttal deadline is approaching, we kindly invite you to join the discussion. We would greatly appreciate it if you could reconsider your rating, provided all your concerns have been addressed. If you have any additional questions, please do not hesitate to let us know. We are more than happy to provide further clarifications.
>
> Thank you again for your careful review and valuable suggestions!
>
> Best,
>
> Authors

---

> > ### Comment · Reviewer_1KdT · 2024-12-02
> >
> > Thank you for clarifying technical detials and conduct additional experiments. I've read all reviews and comment threads, overall I think this is a solid work and can contribute to the community, thus update my evaluation score to 6. I recommend authors to add these abalations (maybe run larger scale experiments with Qwen2VL to get a SOTA model/data) to the paper, and data/code could be open source as promised to further benefit this field.
> >
> > I'd also want to comment on another comment from reviewer about the upper bound of distallation from stronger VLMs. If we look at current LLMs training (distallate knowledge from the entire noisy web data), synthetic data has shown to be very effective. Peronally I think we should explore more synthetic data generation methods for VLMs as well.

---

> ### Author Response · Authors · 2024-12-02
>
> Dear Reviewer 1KdT,
>
> Thank you for taking the time to review our responses and for updating your rating! We sincerely appreciate your recognition of our efforts!  We strongly agree that more exploration of synthetic data for VLMs is needed. The lack of synthetic data, especially data with diversity and complexity, seriously hinders the further improvement of VLM performance. Besides, we will add the ablations to the final version of this paper.
>
> Thank you once again for your valuable suggestions and constructive feedback throughout the entire review process!
>
> Best,
>
> Authors

---

### Public Comment · ~Lai_Wei7 · 2024-11-19
**Thanks for your interesting work**

Have you open-sourced the code of MM-Evol? This is a very useful method and I want to try it in other domains and tasks. Thank you :)

---

### Author Response · Authors · 2024-11-23

We express our gratitude for the insightful comments and constructive feedback from the reviewers on our manuscript. We are pleased to have received positive evaluations from the majority of the reviewers. Moreover, we are delighted to learn that the reviewers recognized the significance of the research problem and the novelty of the core idea (Reviewers 1KdT, pN7Z, and h9M6), as well as the convincing nature of the experiments (Reviewers 1KdT, pN7Z, 8Cfc, h9M6, and u5c1). Based on the reviews, we provide both a general response to the points raised by multiple reviewers and individual responses below to address each reviewer's concerns.

1. Regarding the questions about the experiments, we have taken the following actions:
   - For Reviewers 1KdT, pN7Z, 8Cfc, h9M6, and u5c1, we have either highlighted the locations of the required experiments corresponding to their comments in our paper or added the pertinent experiments accordingly.
   - For Reviewer 1KdT, we have provided an ablation study of the significance of this rewriting step.
   - For Reviewer 1KdT, we have provided more results on OCRBench.
   - For Reviewer 1KdT, we have included ablation experiments with different ratios and information on the elimination rates after three rounds of evolution.
   - For Reviewers 8Cfc and h9M6, we have conducted ablation experiments on different MLLM evolutions, demonstrating the robustness and scalability of our method.
   - For Reviewer 8Cfc, we have provided ablation experiments on different prompt evolutions, showcasing the contribution of our prompt design.
   - For Reviewers 8Cfc and pN7Z, we have supplied additional evaluations showing the consistency of expert evaluations and MLMM evolution scores, further illustrating the reliability of our method.
   - For Reviewers pN7Z and h9M6, we conducted fair comparison experiments under equivalent architecture and data conditions.
   - For Reviewer u5c1, we have provided comparative experimental results between MIMIC-IT and MMEvol, further elucidating the effectiveness and core contributions of our approach.
   - For Reviewer u5c1, we present ablation results on seed data quality, further demonstrating the efficiency and robustness of our method.
2. We have addressed the questions about the idea and technical details as follows:
   - For Reviewer 1KdT, we further explained our technical details and added corresponding case explanations, providing insights regarding data quality and training methods.
   - For Reviewer 8Cfc, we elaborated on the reasons for MMEvol's limited improvements on certain key benchmarks.
   - For Reviewer pN7Z, we further analyzed the potential impact of cumulative error and added more visual cases as suggested.
   - For Reviewer h9M6, we further explained MMEvol's significant potential under fair comparison conditions, and discussed what types of data have better evolutionary potential.
   - For Reviewer u5c1, we further explained the differences between MMEvol and previous methods regarding data diversity and complexity, and method scalability, providing experimental evidence for the efficiency and technical contributions of our approach.
   - For Reviewer u5c1, we supplemented experiments related to the Qwen2 baseline and further explained technical details.
3. Missing reference:
   - For Reviewer 1KdT, we have included the performance data for InternVL2-8b (released 2024/07) and Qwen2-VL-7b (released 2024/08) in the related work section in the revised draft.

We have also revised the draft according to all the reviewers' suggestions, with the revisions highlighted in red. We sincerely thank all the reviewers for their constructive suggestions. Please feel free to let us know if further details or explanations would be useful.

Yours sincerely,
 Authors of #643

---

### Meta-Review · Area_Chair_K66o · 2024-12-22

**Metareview:**

This paper introduces a framework to improve multimodal models by evolving image-text instruction data. It shows some performance improvements with less data. The approach is interesting and shows promising results. However, concerns were raised about the comparison with other methods, and there’s not enough exploration of failure cases or scalability. At this stage, the paper lacks strong support for acceptance, and the authors are encouraged to revise their work, providing more clarity in their experiments.

**Additional Comments On Reviewer Discussion:**

During the rebuttal, reviewers raised concerns about fairness in comparisons with other methods, the clarity of the framework's technical details, and the lack of failure case exploration. While the authors made revisions, including more experiments and clarifications, the overall improvements were marginal, and scalability concerns remained. These points contributed to the decision to reject the paper.

---

### Decision · Program_Chairs · 2025-01-22

Reject